# ✄FlowCut: Rethinking Redundancy via Information Flow for Efficient Vision-Language Models

**Jintao Tong**[1]   **Wenwei Jin**[2‡]   **Pengda Qin**[2]   **Anqi Li**[3]   **Yixiong Zou**[1*]
**Yuhong Li**[2*]   **Yuhua Li**[1]   **Ruixuan Li**[1]

[1]School of Computer Science and Technology, Huazhong University of Science and Technology
[2]Xiaohongshu Inc.   [3]Shanghai Jiao Tong University
[1]{jintaotong, yixiongz}@hust.edu.cn   [2]{wenwei1217.jin, daniel.yuhong}@gmail.com

## Abstract

Large vision-language models (LVLMs) excel at multimodal understanding but suffer from high computational costs due to redundant vision tokens. Existing pruning methods typically rely on single-layer attention scores to rank and prune redundant visual tokens to solve this inefficiency. However, as the interaction between tokens and layers is complicated, this raises a basic question: *Is such a simple single-layer criterion sufficient to identify redundancy?* To answer this question, we rethink the emergence of redundant visual tokens from a fundamental perspective: information flow, which models the interaction between tokens and layers by capturing how information moves between tokens across layers. We find (1) the CLS token acts as an information relay, which can simplify the complicated flow analysis; (2) the redundancy emerges progressively and dynamically via layer-wise attention concentration; and (3) relying solely on attention scores from single layers can lead to contradictory redundancy identification. Based on this, we propose FlowCut, an information-flow-aware pruning framework, mitigating the insufficiency of the current criterion for identifying redundant tokens and better aligning with the model's inherent behaviors. Extensive experiments show that FlowCut achieves superior results, outperforming SoTA by 1.6% on LLaVA-1.5-7B with 88.9% token reduction, and by 4.3% on LLaVA-NeXT-7B with 94.4% reduction, delivering $3.2\times$ speed-up in the prefilling stage. Our code is available at https://github.com/TungChintao/FlowCut.

## 1  Introduction

Large vision-language models (LVLMs) [4, 26, 41, 53] have achieved remarkable progress, demonstrating impressive abilities in multimodal perception and reasoning by effectively bridging visual and linguistic modalities. However, LVLMs face significant challenges in computational costs and memory demands, primarily caused by the massive amounts of vision tokens, especially for high-resolution images [39, 55] and multi-frame videos [36, 44], limiting their practical applications.

As visual signals inherently contain more spatial redundancy than information-dense text [12, 45, 52], a surge of recent efforts [6, 57, 59, 62] introduce token reduction strategies to solve the inefficiency of LVLMs by pruning visual tokens in a training-free manner. These methods typically rely on attention scores to rank and prune tokens either at a specific layer or several LLM layers with a pre-defined prune ratio for each layer. However, regardless of the pruning schedule, redundancy is assessed solely based on the attention scores from the current layer. This raises a basic question: *Are redundancy defined through such a simple single-layer single-criterion score, as the core of current pruning methods, rigorous enough to support pruning decisions?*

---

*Corresponding author. ‡Project leader.

39th Conference on Neural Information Processing Systems (NeurIPS 2025).

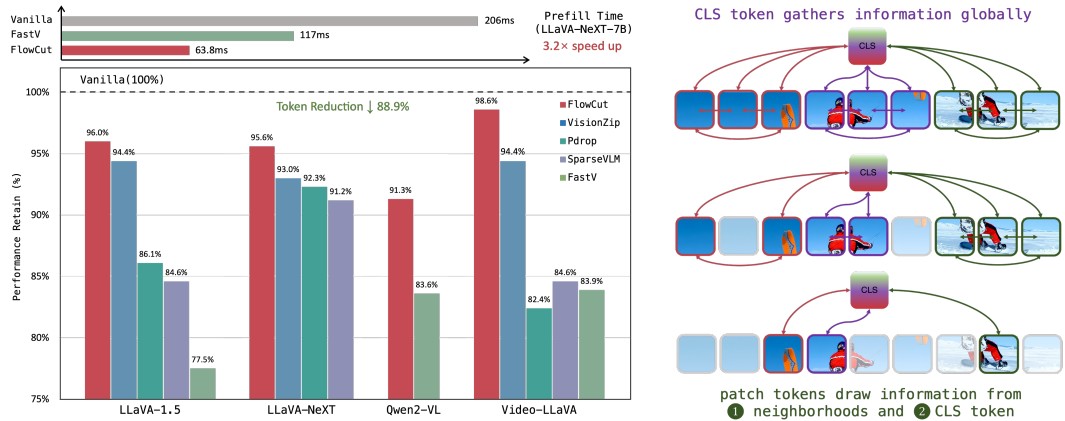

Figure 1: (Left) Performance across various visual understanding benchmarks on diverse LVLMs, FlowCut significantly outperforms other methods. (Right) We provide a unified, bottom-up perspective based on information flow by modeling inter-token interactions across layers, revealing the dynamic emergence of redundancy and guiding pruning decisions to align with this inherent behavior.

In this work, we rethink the emergence of redundant visual tokens from a fundamental perspective: information flow. Specifically, the information flow for each token is defined as the inflow (the source token that transmits the most information to this token) and outflow (the destination token that this token transmits the most information to), which is reflected in the attention map in each layer of ViT and fundamentally models the interaction between tokens and layers. Through systematic analysis of information flow patterns, we uncover a set of key insights that reveal how redundancy emerges, guiding the identification of important tokens: ❶ *The CLS token acts as an information relay and its behavior can largely represent overall information flow*, allowing the complex token-to-token interactions to be approximated and validating it as an effective proxy for identifying redundant tokens. ❷ *Redundancy emerges progressively via layer-wise attention concentration*, suggesting a dynamic pruning strategy aligned with the natural evolution of attention to match the tokens' inherent behavior. ❸ *Relying solely on attention score from single layers can lead to contradictory redundancy identification*, indicating the limitation of single-criterion scoring and single-layer view, which demonstrates the insufficiency of the current criterion for identifying redundancies.

Driven by these insights, we propose FlowCut, an information-flow-aware pruning framework. Specifically, we introduce a layer-wise adaptive pruning ratio module based on attention entropy, adjusting the pruning strength according to the inherent concentration level of attention distributions. We also design a multi-criteria scoring strategy to jointly consider attention strength, information density, and semantic relevance, providing a more reliable estimation of token importance. Further, we develop a cumulative importance evaluation mechanism that aggregates importance scores over layers, mitigating the insufficiency of the current criterion through continuity across layers. Our designs ensure that pruning decisions align with the inherent dynamic information flow of tokens, preserving critical information while effectively eliminating redundancy (Fig. 1 left).

In summary, our contributions can be listed as

- We provide a unified, bottom-up perspective (*information flow*) by modeling token-to-token interaction to explore the propagation pattern of visual information and understand the emergence of redundant, revealing redundancy as a natural consequence of asymmetric attention.

- Based on a comprehensive analysis of the information flow, we verify both the irrationality and the insufficiency of detecting redundant tokens based on a single layer or a single criterion.

- Based on the analysis, we propose FlowCut, an information-flow-aware pruning framework that better aligns with the model's inherent flow of information, mitigating the insufficiency of the current criterion of identifying redundant tokens.

- Extensive experiments show FlowCut achieves superior results: outperforming SoTA by 1.6% on LLaVA-1.5-7B (96.0% vs 94.4%) with 88.9% token reduction, and by 4.3% on LLaVA-NeXT-7B (91.9% vs 87.6%) with 94.4% reduction while delivering 3.2 × speed-up in prefilling stage.

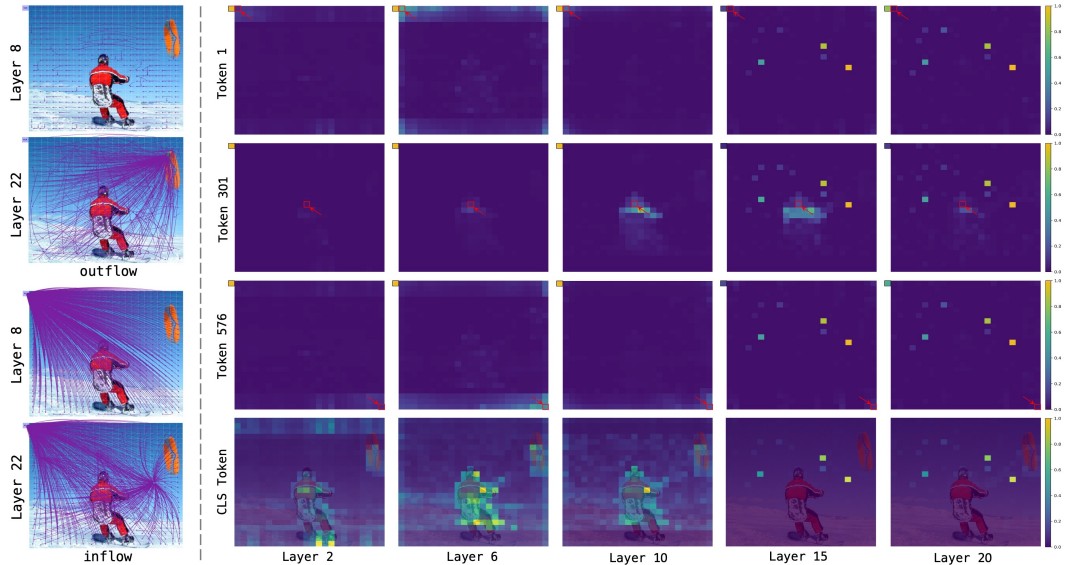

Figure 2: (Left) Information outflow and inflow across vision encoder layers. (Right) Attention map of various tokens across vision encoder layers. Patch tokens engage in sparse, selective information interactions while the CLS token acts as a context relay that gathers and provides information globally.

## 2 Rethinking Visual Redundancy from Information Flow

Information flow offers a unified, bottom-up perspective for understanding token interactions. In this section, we systematically delve into information flow to analyze the emergence of redundancy.

### 2.1 Preliminaries

**Architecture of LVLMs.** The architecture of LVLMs generally comprises three key components: 1) a pre-trained vision encoder, which transforms input images into visual tokens; 2) a modality connector, serving as a bridge between the vision encoder and the LLM by aligning visual tokens with the LLM's word embedding space; and 3) a pre-trained LLM, which integrates the aligned visual and textual information to perform reasoning and generate responses.

**Visual Token Processing.** LVLMs process visual tokens in three stages: vision encoding, prefilling, and decoding. Images are first divided into patches and encoded into visual tokens by a vision backbone. These tokens are then fused with textual tokens during the prefilling stage to form joint representations, while key-value pairs are cached for generation. In the decoding stage, new tokens are generated autoregressively using the cached information, enabling efficient multimodal inference.

### 2.2 Delve into Information Flow of Visual Tokens

#### i) Analysis of Information Flow

We begin by analyzing information inflow and outflow patterns across vision encoder layers. Specifically, for an attention map $\mathbf{A} \in \mathbb{R}^{N \times N}$, we define information inflow for each token as its primary source of information ($\mathrm{argmax}(\mathbf{A}[i,:])$) and outflow as the primary destination of its information ($\mathrm{argmax}(\mathbf{A}[:,j])$). Figure 2 (Left) reveals two distinct information flow patterns: CLS tokens consistently engage in global information exchange while providing information to patch tokens, whereas patch tokens exhibit short-range, local, uniform flows in shallow layers and long-range, global, deterministic flows in deeper layers.

To further understand information flow, we analyze the attention map of both the CLS token and patch tokens across layers [1, 65]. As shown in Figure 2 (Right, rows 1-3), each patch token consistently focuses on a small subset of tokens. While this focus is initially on the CLS token and its neighbors in shallow layers, in deep layers, each patch token uniformly directs substantial attention to a specific group of non-neighboring, non-CLS tokens, which we identify as 'hub tokens' due to

this concentrated attention. In contrast, the CLS token (Figure 2, Right, row 4) initially distributes its attention broadly across most tokens, engaging with a wide spectrum of tokens. As layers get deeper and the 'hub tokens' emerge as the focus for patch tokens, the CLS token's attention pattern also gradually converges towards these hub tokens.

> ***Insight 1*** Information flow offers a unified, bottom-up perspective for understanding token interactions, agnostic to model architectures. It reveals redundancy as a natural consequence of asymmetric attention: as layers deepen, attention concentrates on fewer tokens, and tokens that no one focuses on are naturally the redundant ones.

However, while conceptually powerful, modeling all token-to-token flows across all layers is costly and analytically intractable, necessitating a simplified yet representative proxy.

### ii) The CLS Token Serves as the Relay that Represents the Overall Information Flow

As in Fig. 2 (Right), the attention patterns of both patch and CLS tokens gradually converge to the same hub token in deeper layers. This trend implies an information relay mechanism: for a given patch token, since shallow-layer patch tokens primarily attend only to their local neighbors and the CLS token, it has minimal direct access to information from distant patch tokens; therefore, the CLS token, as it can gather global information from most patch tokens, must have transmitted information of distant patches to the given patch token to direct this patch's attention in deep layers to these distant patch tokens, i.e., the CLS token serves as the relay to transmit information across patch tokens. Moreover, as patch tokens derive information from the CLS token, their attention gradually mirrors its preferences, therefore, their attention converges to the same hub tokens.

Furthermore, we measure the average distance between each token and the token that it attends the most in Fig. 3 (Left), where the attention distance exhibits an overall increasing trend as the layer deepens. This further verifies that patch tokens initially gather information primarily from nearby neighbors and rely on the CLS token for capturing distant semantic context.

> ***Insight 2*** The CLS token acts as a global information relay, broadcasting distant-patch information to each patch token. As layers deepen, patch tokens increasingly mirror the CLS token's focus, converging toward the same set of hub tokens, suggesting we can take the CLS token as a proxy to simplify the complex information flow and identify critical tokens.

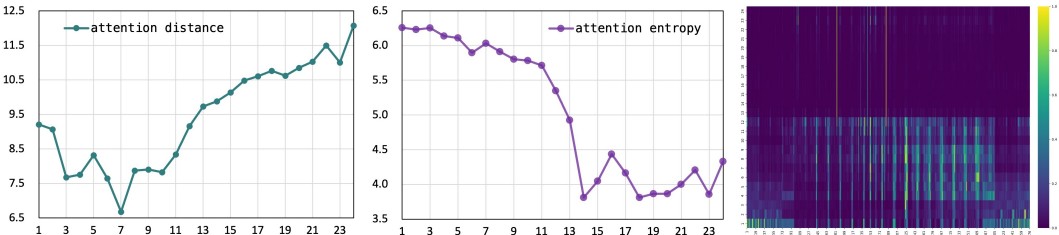

Figure 3: (Left) Average attention distance of patch tokens generally increases as the layer deepens. (Middle) Attention entropy across vision encoder layers. (Right) The CLS token attention across layers: dynamic evolution of attention distributions.

### iii) Progressive Attention Concentration and Emergence of Redundancy

Our above analysis reveals that the CLS token's behavior can largely represent the overall information flow. Therefore, we visualize the overall information flow tendency by measuring the CLS token's attention across the vision encoder layers. Fig. 3 (Right) demonstrates that attention pattern distributions dynamically evolve across layers, and as information traverses deeper layers, attention generally narrows, progressively concentrating on a smaller group of tokens. This increasing concentration of attended tokens inherently implies that other tokens become less critical to the information flow. We interpret this diminishing attention as the progressive emergence of semantic redundancy. Thus, redundancy appears not as a static feature but as an emerging property, developing through complex, layer-by-layer interactions during the encoding process.

To quantify the trend of attention concentration, we measured the entropy of the attention distribution at each layer. As in Fig. 3 (Middle), the attention entropy exhibits an overall decreasing trend.

Notably, a sharp decline in entropy is observed between layers 11 and 15, signifying a period of rapid attention focusing. This result quantitatively supports the progressive concentration of attention, verifying that redundancy naturally emerges as attention becomes more concentrated.

> **Insight 3**   Redundancy emerges progressively as attention concentrates over depth, not abruptly at the end but layer by layer, driven by increasingly focused semantic interactions. This layer-specific development of redundancy suggests that optimal pruning should be dynamic and layer-adaptive rather than applying fixed ratios across layers or pruning only at the final stages.

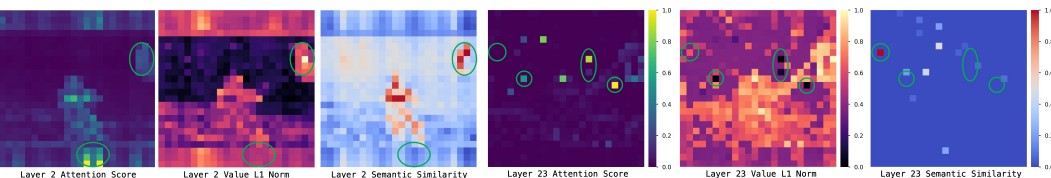

Figure 4: Different criteria for identifying critical tokens can show contradictory results, implying the instability of each single criterion in each single layer.

### iv) Beyond High Attention: Identifying Truly Important Tokens

Given the complex interactions between tokens and layers modeled by information flow, a critical question naturally arises: Is the simple single-layer criterion, as has been adopted by current works, sufficient to capture such a complex generation of redundant tokens? To investigate this issue, we analyze several metrics at different layers of the vision encoder (selected for feeding into LLM). Alongside attention scores, we measured: 1) the L1 norm of the Value vectors, which serves as an indicator of their information density [16, 22], where a small norm suggests a weak signal and thus less information conveyed by Value vectors; and 2) the semantic similarity between each token and the CLS token. Given that the CLS token aggregates global information, this similarity score reflects the token's semantic alignment with, or relevance to, the overall global context [34].

The results presented in Fig. 4 demonstrate that some tokens highly attended to by the CLS token can exhibit remarkably low information density (as measured by the Value L1 norm) and low semantic relevance to the global context (as measured by semantic similarity to the CLS token). This means that applying different criteria, such as the high attention scores of the CLS token, in identifying critical tokens can lead to contradictory results, even though all these criteria are reasonable. This problem may be the result of the noisy information in the information flow. Since the flow is transmitted through every two tokens and every connected layers, every small perturbation (noise) in the token/attention could be amplified to lead to wrong identification of critical tokens.

> **Insight 4**   High attention score from a single layer is not a bad criterion, but is still insufficient in measuring token importance due to the complexity in token interactions. An ideal criterion should take networks' inherent behavior into account and resist the instability of current criteria.

## 3   Methodology

In this section, based on the above analysis, we introduce our method for visual token reduction, which better fits the model's inherent information flow, including flow distribution-aware prune ratio, cumulative flow importance tracking, and multi-criteria evaluator. The overall framework is in Fig. 5.

### 3.1   Attention Distribution-Aware Prune Ratio

Redundancy emerges progressively as attention becomes more concentrated. Based on this insight, since the CLS token acts as the relay of information flow, we adopt the CLS token's attention entropy to adaptively adjust pruning ratios: higher entropy indicates widespread token contribution to information flow, necessitating conservative pruning, while lower entropy reflects concentrated attention and emerging redundancy, permitting more aggressive pruning. Specifically, the attention score with $h$ heads $\mathbf{A} \in \mathbb{R}^{h \times N \times N}$ is computed as:

$$\mathbf{A} = \text{Attention}(\mathbf{Q}, \mathbf{K}) = \text{Softmax}\left(\frac{\mathbf{Q}\mathbf{K}^\top}{\sqrt{D_k}}\right) \tag{1}$$

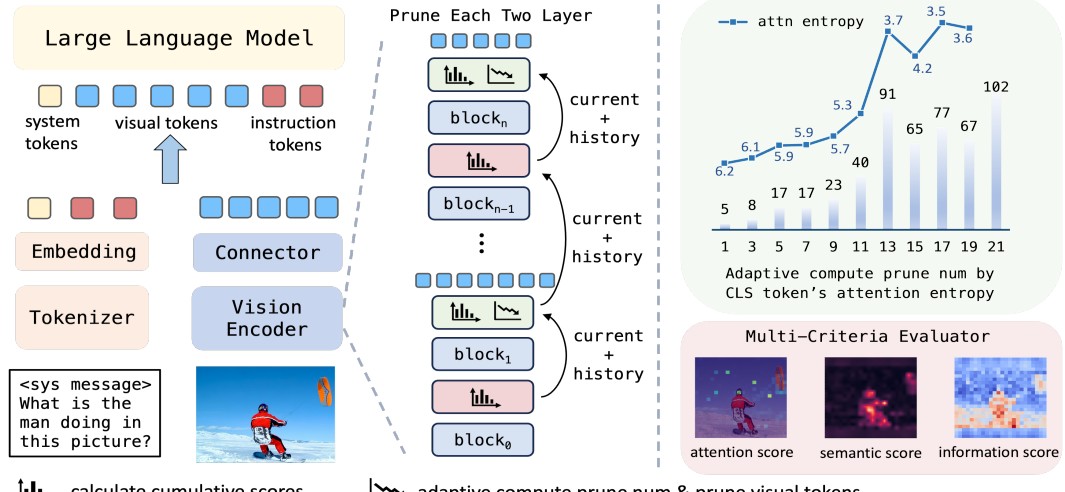

Figure 5: The overview of FlowCut, an information flow-aware pruning framework. The process involves: (1) adaptively determining pruning ratios based on attention concentration; (2) evaluating token importance via multiple criteria; and (3) pruning tokens based on combined current and historical scores, with historical values updated accordingly.

where $\mathbf{Q}$ and $\mathbf{K}$ denote query and key matrices, $D_k$ is the key dimension. By averaging $\mathbf{A}$ across all heads, we obtain an aggregated attention matrix $\mathbf{A}_{\text{avg}} \in \mathbb{R}^{N \times N}$, here $N$ is the sequence length.

For vision encoders with a CLS token, we extract the CLS token's attention map as global attention vector $\mathbf{A}^{\text{g}}$. For those without a CLS token, we first average all tokens to create a global token, then compute attention scores between this global token and all sequence tokens to derive $\mathbf{A}^{\text{g}}$. The global attention vector $\mathbf{A}^{\text{g}} \in \mathbb{R}^{1 \times N}$ is then used to compute the attention entropy $H(\mathbf{A}^{\text{g}})$ as follow:

$$H_{\max} = \log N, \ \ H(\mathbf{A}^{\text{g}}) = -\sum_{i=1}^{N} A_i^g \log A_i^g \tag{2}$$

where, $H_{max}$ denotes the maximum possible entropy when attention is uniformly distributed. The final visual tokens' prune number $P$ is computed as:

$$P = \left(\frac{N - T}{\sqrt{L}}\right) \cdot \left(1 - r_H^2\right), \ \text{where} \ \ r_H = \frac{H(\mathbf{A}^{\text{g}})}{H_{\max}} \tag{3}$$

where $N$ is the current visual token number, $T$ is the target number, $L$ is the remain prune layers, and $P$ will be rounded to the nearest integer by $\text{round}(P)$.

## 3.2 Multi-Criteria Evaluator

Existing methods typically rely on a single criterion, such as the attention score of the CLS token or the last instruction token, which is verified by us to be insufficient to accurately capture the role of each token in information flow. To address this limitation, we propose a multi-criteria evaluation framework that incorporates attention strength, semantic similarity, and information density.

Specifically, for attention strength $\mathbf{I}^{\text{a}}$, we use $\mathbf{A}^{\text{g}}$ as the metric. For semantic similarity $\mathbf{I}^{\text{s}}$, we establish a global value vector $\mathbf{V}^{\text{g}}$ by using either the CLS token's value vector (for encoders with a CLS token) or the average of all tokens' value vectors (for encoders without a CLS token), and then calculate the similarity between $\mathbf{V}^{\text{g}}$ and patch token's value vector. For information density $\mathbf{I}^{\text{d}}$, we measure it by computing the L1 norm of patch token's value vector. The formula as follow:

$$\mathbf{I}^{\text{a}} = \mathbf{A}^{\text{g}}, \ \ \mathbf{I}^{\text{s}} = \text{Softmax}\left(\frac{\mathbf{V_g V}^{\top}}{\sqrt{D_v}}\right), \ \ \mathbf{I}^{\text{d}} = \|\mathbf{V}\|_1 \tag{4}$$

where $D_v$ is the dimension of $\mathbf{V}$. The final importance score $\mathbf{S}$ is caculate as:

$$\mathbf{S} = \left(\frac{\mathbf{I}^{\text{a}}}{\sum_{k=1}^{N} I_k^a} + \frac{\mathbf{I}^{\text{s}}}{\sum_{k=1}^{N} I_k^s}\right) \times \mathbf{I}^{\text{d}} \tag{5}$$

Table 1: **Performance of FlowCut on LLaVA-1.5 under different visual token settings.** The vanilla number of visual tokens is 576. The last column is the average accuracy proportion relative to the upper limit. FlowCut achieves optimal performance on various image understanding benchmarks. The comparison with additional methods and setting is provided in Appendix A.3.

| Method | GQA | MMB | MMBCN | MME | POPE | SQA | VQAV2 | VQAText | VizWiz | SEED | MMVet | LLaVA-B | Avg. |
|---|---|---|---|---|---|---|---|---|---|---|---|---|---|
| *Upper Bound, 576 Tokens* **(100%)** | | | | | | | | | | | | | |
| Vanilla | 61.9 | 64.7 | 58.1 | 1862 | 85.9 | 69.5 | 78.5 | 58.2 | 50.0 | 59.3 | 31.1 | 66.9 | 100% |
| *Retain 192 Tokens* (↓ **66.7%**) | | | | | | | | | | | | | |
| ToMe (ICLR23) | 54.3 | 60.5 | - | 1563 | 72.4 | 65.2 | 68.0 | 52.1 | - | 53.1 | 27.9 | - | 88.7% |
| FastV (ECCV24) | 52.7 | 61.2 | 57.0 | 1612 | 64.8 | 67.3 | 67.1 | 52.5 | 50.8 | 57.1 | 27.7 | 49.4 | 89.4% |
| SparseVLM (ICML25) | 57.6 | 62.5 | 53.7 | 1721 | 83.6 | **68.9** | 75.6 | 56.1 | 50.5 | 55.8 | 31.5 | 66.1 | 96.6% |
| PDrop (CVPR25) | 57.3 | 62.9 | 56.8 | 1766 | 82.3 | 68.8 | 75.1 | 56.5 | 51.1 | 54.7 | 30.5 | 65.8 | 96.7% |
| VisionZip (CVPR25) | 59.3 | 63.0 | 57.3 | 1778 | 85.2 | 68.7 | 76.6 | 57.3 | 51.2 | 56.4 | 31.7 | 67.7 | 98.5% |
| **FlowCut (Ours)** | **60.1** | **63.2** | **57.8** | **1836** | **86.1** | 68.6 | **77.1** | **57.5** | **51.8** | 57.5 | **34.3** | **68.4** | **100.2%** |
| *Retain 128 Tokens* (↓ **77.8%**) | | | | | | | | | | | | | |
| ToMe (ICLR23) | 52.4 | 53.3 | - | 1343 | 62.8 | 59.6 | 63.0 | 49.1 | - | 50.9 | 27.2 | - | 81.8% |
| FastV (ECCV24) | 49.6 | 56.1 | 56.4 | 1490 | 59.6 | 60.2 | 61.8 | 50.6 | 51.3 | 55.9 | 28.1 | 52.0 | 85.9% |
| SparseVLM (ICML25) | 56.0 | 60.0 | 51.1 | 1696 | 80.5 | 67.1 | 73.8 | 54.9 | 51.4 | 53.4 | 30.0 | 62.7 | 93.8% |
| PDrop (CVPR25) | 57.1 | 61.6 | 56.3 | 1664 | 82.3 | 68.3 | 72.9 | 56.6 | 51.0 | 53.3 | 30.8 | 61.9 | 95.1% |
| VisionZip (CVPR25) | 57.6 | 62.0 | 56.2 | 1759 | 83.2 | **68.9** | 75.6 | 56.8 | 51.6 | 54.9 | 32.1 | 64.8 | 97.2% |
| **FlowCut (Ours)** | **58.5** | **62.1** | **56.5** | **1792** | **85.2** | 68.6 | **76.0** | **57.3** | **52.2** | **56.2** | **32.5** | **67.5** | **98.5%** |
| *Retain 64 Tokens* (↓ **88.9%**) | | | | | | | | | | | | | |
| ToMe (ICLR23) | 48.6 | 43.7 | - | 1138 | 52.5 | 50.0 | 57.1 | 45.3 | - | 44.0 | 24.1 | - | 71.4% |
| FastV (ECCV24) | 46.1 | 48.0 | 52.7 | 1256 | 48.0 | 51.1 | 55.0 | 47.8 | 50.8 | 51.9 | 25.8 | 46.1 | 77.5% |
| SparseVLM (ICML25) | 52.7 | 56.2 | 46.1 | 1505 | 75.1 | 62.2 | 68.2 | 51.8 | 50.1 | 51.1 | 23.3 | 57.5 | 84.6% |
| PDrop (CVPR25) | 47.5 | 58.8 | 50.5 | 1561 | 55.9 | 68.6 | 69.2 | 50.6 | 50.7 | 40.0 | 30.7 | 59.2 | 86.1% |
| VisionZip (CVPR25) | 55.1 | 60.1 | 55.3 | 1687 | 77.0 | 69.0 | 72.4 | 55.5 | 52.8 | 52.2 | 31.5 | 62.9 | 94.4% |
| **FlowCut (Ours)** | **55.6** | **60.8** | **55.4** | **1744** | **80.2** | **69.1** | **72.8** | **55.6** | **53.2** | **53.5** | **32.3** | **65.1** | **96.0%** |

## 3.3 Cumulative Flow Importance Tracking

Based on the observation that attention distribution dynamically and continually shift across layers, and the single-layer criterion is insufficient to capture a token's importance, we track cumulative importance by aggregating both historical and current criteria across layers, enabling a more accurate and stable assessment of each token's overall contribution. Specifically, we gain multi-criteria score $\mathbf{S}_{cur}$ each layer and update cumulative score $\mathbf{S}_{cum}$ using both the current and historical values:

$$\mathbf{S}_{cum}^{(l)} = 0.5 \times \mathbf{I}_{cur}^{(l)} + 0.5 \times \mathbf{S}_{cum}^{(l-1)}, \quad \text{where} \quad \mathbf{S}_{cum}^{(0)} = \mathbf{I}_{cur}^{(0)} \quad \text{if} \ layer = 0 \tag{6}$$

We prune every two layers, relying on the cumulative score to ensure that the importance value takes into account both current and historical values, which implicitly filters out noise in flows.

## 4 Experiments

**Experiment Setting.** To demonstrate FlowCut's effectiveness, we conducted extensive experiments on diverse open-source LVLMs. For image understanding tasks, we implemented our designs on the widely-used LLaVA family—specifically LLaVA-1.5-7B [39] and the high-resolution LLaVA-Next-7B [40](supporting up to 2880 image tokens)—evaluating across twelve benchmarks with varying reduction ratios. To further validate the universality of our method, we extended evaluation to the more advanced Qwen2-VL [55] model. For video understanding tasks, we employed Video-LLaVA [36] as a baseline and verified our method across three video benchmarks.

**Implement Details** Our method can be seamlessly integrated into models in a training-free manner, or training from scratch, resulting in substantial improvements to inference speed and training efficiency. For LLaVA-1.5, we complete the pruning at the penultimate layer of the vision encoder. For LLaVA-NeXT and Qwen2-VL, which contain a larger number of tokens, our pruning process spans from the vision encoder to the second layer of the LLM, which we prune tokens to twice the target number in the vision encoder, then further reduce to the final target number at the second layer of the LLM. All of our experiments are conducted on Nvidia A800-80G GPU.

Table 3: Performance of FlowCut on Qwen2-VL-7B for image understanding. As the original token numbers is dynamical, the reduction ratio is approximate.

| Method | GQA | MMB | MMB$^{CN}$ | MME | POPE | SQA | VQA$^{Text}$ | Avg. |
|---|---|---|---|---|---|---|---|---|
| *Upper Bound, All Tokens* **(100%)** | | | | | | | | |
| Vanilla | 61.9 | 79.9 | 79.5 | 2338 | 87.2 | 85.1 | 82.2 | 100% |
| *Token Reduction* (↓ **66.7%**) | | | | | | | | |
| FastV | 58.0 | 76.1 | 73.9 | 2130 | 82.1 | 80.0 | 77.3 | 93.6% |
| **FlowCut** | 60.5 | 79.2 | 78.2 | 2335 | 86.0 | 84.0 | 81.1 | **98.7%** |
| *Token Reduction* (↓ **77.8%**) | | | | | | | | |
| FastV | 56.7 | 74.1 | 72.4 | 2031 | 79.2 | 78.3 | 72.0 | 90.4% |
| **FlowCut** | 59.2 | 77.8 | 76.9 | 2310 | 84.6 | 80.5 | 78.3 | **96.5%** |
| *Token Reduction* (↓ **88.9%**) | | | | | | | | |
| FastV | 51.9 | 70.1 | 63.8 | 1962 | 76.1 | 75.8 | 60.3 | 83.6% |
| **FlowCut** | 56.4 | 72.6 | 72.5 | 2252 | 81.8 | 78.2 | 68.9 | **91.3%** |

Table 4: Performance of FlowCut on video understanding tasks. The original Video-LLaVA's video token number is 2048, while we only retain the 256 tokens.

| Method | TGIF Acc | TGIF Score | MSVD Acc | MSVD Score | MSRVT Acc | MSRVT Score | Avg. Acc | Avg. Score |
|---|---|---|---|---|---|---|---|---|
| Video-LLaVA | 46.9 | 3.34 | 69.8 | 3.91 | 57.1 | 3.49 | 100% | 100% |
| FastV | 44.2 | 3.29 | 60.3 | 3.72 | 40.6 | 3.18 | 83.9% | 94.9% |
|  | 94.2% | 98.5% | 86.4% | 95.1% | 77.1% | 91.1% | | |
| SparseVLM | 45.9 | 3.32 | 68.6 | 3.90 | 32.9 | 3.02 | 84.6% | 95.2% |
|  | 98.9% | 99.4% | 98.3% | 99.7% | 57.6% | 86.5% | | |
| PDrop | 40.3 | 3.21 | 61.5 | 3.74 | 41.8 | 3.19 | 82.4% | 94.4% |
|  | 85.9% | 96.1% | 88.1% | 95.7% | 73.2% | 91.4% | | |
| VisionZip | 44.3 | 3.29 | 65.2 | 3.83 | 54.5 | 3.43 | 94.4% | 98.2% |
|  | 94.5% | 98.5% | 93.4% | 98.0% | 95.4% | 98.3% | | |
| **FlowCut** | 46.8 | 3.35 | 68.8 | 3.91 | 55.7 | 3.46 | **98.6%** | **99.8%** |
|  | 99.8% | 100.3% | 98.6% | 100% | 97.5% | 99.1% | | |

## 4.1 Effectiveness of FlowCut in Inference

**Image understanding tasks** We apply FlowCut to LLaVA-1.5-7B during inference without additional training and compare it with other approaches. The results in Table 1 demonstrate that FlowCut achieves optimal results on almost all benchmarks, with its average performance significantly exceeding all existing methods. we conduct further experiments on LLaVA-NeXT-7B. As shown in Table 2, FlowCut outperforming other methods by a significant margin. We even surpassing VisionZip by 4.3% while retaining only 160 tokens. Furthermore, as shown in Table 3, we extended the evaluation to Qwen2-VL to validate that FlowCut can be applied seamlessly to different architectures.

Table 2: Comparison of FlowCut on LLaVA-NeXT-7B with other methods.

| Method | GQA | MMB | MMB$^{CN}$ | MME | POPE | VQA$^{V2}$ | VQA$^{Text}$ | SEED | Avg. |
|---|---|---|---|---|---|---|---|---|---|
| *Upper Bound, 2880 Tokens* **(100%)** | | | | | | | | | |
| Vanilla | 64.2 | 67.9 | 60.6 | 1846 | 86.4 | 81.8 | 61.3 | 63.6 | 100% |
| *Retain 640 Tokens* (↓ **77.8%**) | | | | | | | | | |
| SparseVLM | 60.3 | 65.8 | 58.5 | 1773 | 84.2 | 77.1 | 57.8 | 59.3 | 95.3% |
| PDrop | 60.6 | 65.5 | 58.5 | 1781 | 83.7 | 78.3 | 57.4 | 60.5 | 95.7% |
| VisionZip | 61.3 | 66.2 | 57.8 | 1787 | 85.9 | 79.1 | 60.2 | 60.9 | 96.9% |
| **FlowCut** | 61.9 | 66.7 | 59.2 | 1837 | 86.1 | 79.8 | 60.6 | 61.9 | **98.2%** |
| *Retain 320 Tokens* (↓ **88.9%**) | | | | | | | | | |
| SparseVLM | 57.7 | 63.2 | 54.4 | 1685 | 82.2 | 73.4 | 55.9 | 56.9 | 91.2% |
| PDrop | 58.3 | 63.9 | 56.8 | 1736 | 80.2 | 75.2 | 55.3 | 57.8 | 92.3% |
| VisionZip | 59.2 | 63.1 | 55.3 | 1702 | 82.1 | 76.2 | 58.9 | 58.2 | 93.0% |
| **FlowCut** | 59.8 | 65.3 | 57.5 | 1791 | 83.4 | 77.8 | 60.1 | 59.5 | **95.6%** |
| *Retain 160 Tokens* (↓ **94.4%**) | | | | | | | | | |
| SparseVLM | 51.2 | 52.1 | 48.6 | 1542 | 72.7 | 66.3 | 46.4 | 49.2 | 79.8% |
| PDrop | 54.9 | 61.8 | 54.9 | 1513 | 72.3 | 70.2 | 52.7 | 53.9 | 86.2% |
| VisionZip | 55.5 | 60.1 | 52.7 | 1628 | 74.8 | 71.4 | 56.2 | 54.2 | 87.6% |
| **FlowCut** | 57.6 | 62.8 | 55.2 | 1746 | 79.9 | 74.6 | 57.6 | 56.8 | **91.9%** |

**Video understanding tasks** To assess our method's effectiveness in video understanding, we applied it to Video-LLaVA, which encodes each video into 8 frames with 256 visual tokens per frame (totaling 2048 visual tokens). We compared FlowCut to other methods under the setting of retaining only 256 tokens in total (32 per frame). As shown in Table 4, FlowCut achieves 98.6% accuracy across three benchmarks, outperforming the previous state-of-the-art VisionZip by 4.2%. This notable gain highlights it's superior reasoning capability in handling complex multimodal video data.

**Efficiency Analysis** Our proposed FlowCut substantially improves inference efficiency in LVLMs while maintaining comparable performance. As shown in Table 5, we compare total inference time, prefilling time, and FLOPs on both LLaVA-1.5-7B and LLaVA-NeXT-7B against several existing methods. FlowCut consistently achieves the best efficiency gains. Notably, it delivers a **3.2× speedup** in prefilling and a **3.0× speedup** in the entire inference on LLaVA-NeXT.

Table 5: **Efficiency analysis of FlowCut on LLaVA-1.5-7B (left) and LLaVA-NeXT-7B (right).** The detailed metrics include practical total running time (min:sec), prefilling time and TFLOPs. $\Delta$ denotes the inference speedup. The results are tested for one A800 GPU on POPE benchmark.

| Methods | Token | Total Time↓ | Δ↑ | Prefilling Time↓ | Δ↑ | TFLOPs |
|---|---|---|---|---|---|---|
| LLaVA-1.5-7B | 576 | 17:05 | 1.0× | 102ms | 1.0× | 8.82 |
| + FastV | 64 | 15:32 | 1.1× | 87.3ms | 1.2× | 2.26 |
| + SparseVLM | 64 | 15:57 | 1.1× | 90.1ms | 1.1× | 2.31 |
| + PDrop | 64 | 12:56 | 1.3× | 72.5ms | 1.4× | 2.16 |
| + VisionZip | 64 | 12:10 | 1.4× | 69.2ms | 1.5× | 2.03 |
| **+ FlowCut** | 64 | **11:17** | **1.5×** | **62.6ms** | **1.6×** | **1.95** |

| Methods | Token | Total Time↓ | Δ↑ | Prefilling Time↓ | Δ↑ | TFLOPs |
|---|---|---|---|---|---|---|
| LLaVA-NeXT-7B | 2880 | 35:16 | 1.0× | 206ms | 1.0× | 31.03 |
| + FastV | 160 | 28:27 | 1.3× | 117ms | 1.8× | 7.35 |
| + SparseVLM | 160 | 30:26 | 1.2× | 136ms | 1.5× | 7.62 |
| + PDrop | 160 | 13:45 | 2.6× | 79.5ms | 2.6× | 6.78 |
| + VisionZip | 160 | 12:32 | 2.8× | 69.9ms | 2.9× | 4.72 |
| **+ FlowCut** | 160 | **11:40** | **3.0×** | **63.8ms** | **3.2×** | **4.29** |

## 4.2 Effectiveness of FlowCut in Training

Our method can also be seamlessly integrated into models during training to reduce the number of visual tokens, thereby saving memory and shortening training time. As shown in Table 6, we evaluate FlowCut with different token configurations across 10 benchmarks on LLaVA-1.5-7B. The results demonstrate that with 192 tokens, FlowCut achieves a $1.5\times$ training speedup while actually improving performance to 101.1% of the original model. Even with further reduction to 128 tokens, training speedup reaches $1.6\times$ while still retaining 99.8% of the original performance.

Table 6: **Performance and Training Time of FlowCut on LLaVA-1.5-7B.** The first line of each method shows the raw benchmark accuracy, and the second line is the proportion relative to the upper limit. The time refers to the practical training duration (pretrain time + fine-tuning time), while $\Delta$ indicates training acceleration factor. All experiments are conducted on 8 Nvidia A800 GPUs.

| Method | Tokens | Time↓ | $\Delta$↑ | GQA | MMB | MMB$^{\text{CN}}$ | MME | POPE | SQA | VQA$^{\text{V2}}$ | VQA$^{\text{Text}}$ | SEED | MMVet | Avg. |
|---|---|---|---|---|---|---|---|---|---|---|---|---|---|---|
| LLaVA-1.5 7B | 576 | 12.97h | 1.0× | 61.9 | 64.7 | 58.1 | 1862 | 85.9 | 69.5 | 78.5 | 58.2 | 59.3 | 31.1 | 100% |
| FlowCut | 192 | 8.65h | 1.5× | 62.0 | 66.4 | 59.2 | 1840 | 86.0 | 68.9 | 78.7 | 58.1 | 59.2 | 33.6 | 101.1% |
| | | | | 100.2% | 102.6% | 101.9% | 98.8% | 100.1% | 99.1% | 100.3% | 99.8% | 99.8% | 108.0% | |
| FlowCut | 128 | 7.98h | 1.6× | 60.8 | 66.9 | 59.5 | 1835 | 84.5 | 69.2 | 77.9 | 57.9 | 58.6 | 31.2 | 99.8% |
| | | | | 98.2% | 103.4% | 102.4% | 98.5% | 98.4% | 99.6% | 99.2% | 99.5% | 98.8% | 100.3% | |

## 4.3 Ablation Studies

**Effectiveness of each design** To assess the contribution of each design in FlowCut, we conduct ablation studies on three benchmarks under a setting that retains only 128 visual tokens (↓77.8%), as shown in Table 7. Naively pruning at a single layer leads to significant performance drops. Introducing either the cumulative importance evaluation or the multi-criteria scoring strategy individually brings consistent gains across benchmarks, confirming their effectiveness. Moving from single-layer to multi-layer pruning yields notable improvements, yet applying a uniform prune ratio across layers remains suboptimal. Enabling adaptive prune ratio notably enhances performance (e.g., +1.8% on POPE). The best results are achieved when all components are integrated in FlowCut, yielding minimal performance degradation compared to the unpruned baseline. These results verify that each component contributes independently and synergistically to performance.

**Adaptive layer-wise pruning outperforms single-layer pruning** As shown in Figure 6, we measure information density across layers by the L1 norm of value vectors, where lower norms indicate sparser and less informative representations [22]. In single-layer pruning, the information density drops sharply after pruning, indicating substantial information loss. In contrast, FlowCut aligns pruning decisions with the token's dynamic inherent information flow, effectively preserving essential content.

Table 7: Ablation study of various designs with 128 tokens retain across three benchmarks.

| Methods | Token | Adaptive Prune Num | Cumulative Evaluate | Multiple Criterion | TextVQA | GQA | POPE |
|---|---|---|---|---|---|---|---|
| LLaVA-1.5-7B | 576 | – | – | – | 58.2 | 61.9 | 85.9 |
| Single-Layer | 128 | – | | | 55.3 | 54.9 | 76.6 |
| Single-Layer | 128 | – | ✓ | | 56.2 | 57.6 | 83.7 |
| Single-Layer | 128 | – | | ✓ | 55.9 | 57.3 | 82.0 |
| Single-Layer | 128 | – | ✓ | ✓ | 56.8 | 57.8 | 83.9 |
| Multi-Layer | 128 | | | | 56.2 | 56.3 | 81.5 |
| Multi-Layer | 128 | ✓ | | | 57.1 | 58.0 | 83.3 |
| Multi-Layer | 128 | | ✓ | | 56.8 | 58.2 | 83.8 |
| Multi-Layer | 128 | | | ✓ | 56.7 | 57.8 | 83.5 |
| Multi-Layer | 128 | ✓ | ✓ | | 57.0 | 58.3 | 84.0 |
| Multi-Layer | 128 | ✓ | | ✓ | 57.1 | 58.1 | 83.8 |
| Multi-Layer | 128 | | ✓ | ✓ | 56.9 | 57.9 | 84.3 |
| **FlowCut** | 128 | ✓ | ✓ | ✓ | **57.3** | **58.5** | **85.2** |

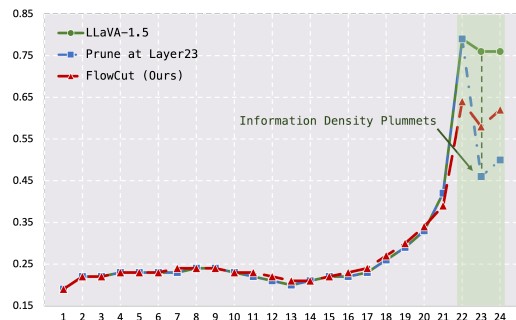

Figure 6: Information density measured by Value vector's L1 norm shows that layer-wise progressive pruning results in less information loss.

## 4.4 Analysis and Discussion

**FlowCut effectively preserves critical information.** According to the attention computation, larger value vectors (V) contribute more to the output, indicating richer information content [17]. We use

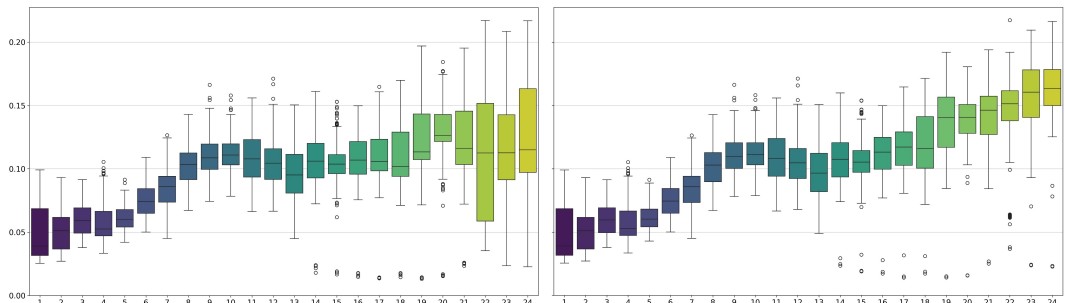

Figure 7: Box plot of Value vector (V) across layers before (left) and after (right) adopting FlowCut.

box plots (Figure 7) to compare the distribution of value vectors before and after applying FlowCut. After pruning, the distribution shifts toward higher magnitudes, suggesting that although fewer tokens remain, each retains more information, effectively preserving overall content richness.

**FlowCut guides model focus more effectively.** Figure 8 compares the pruning results and VQA answers of FlowCut and the previous SoTA. VisionZip relies on single-layer attention scores, resulting in most tokens retained after pruning being concentrated in the center of the image. In contrast, FlowCut leverages multi-layer and multi-criteria evaluations, leading to a more uniform spatial distribution of retained tokens. Moreover, by effectively eliminating redundant and distracting tokens, FlowCut produces more accurate answers than even the non-pruned baseline in some cases.

## 5   Related Work

**Visual Token Compression**   In LVLMs, visual tokens typically contain substantial redundant information. Various methods [5, 32, 55] have been proposed to address this inefficiency. FastV [6] leverages attention scores at the LLM's second layer for pruning, while SparseVLM [62] introduces text-guided pruning through cross-modal attention. VisionZip [59] employs CLS token attention at the vision encoder's final layer for compression, and PDrop [57] uses the last instruction token's attention map to drop visual tokens at several pre-selected layers according to manually designed fixed pruning ratios. However, these methods evaluate token importance using attention scores from a single layer, limiting their comprehensiveness. Our method, instead, accu-

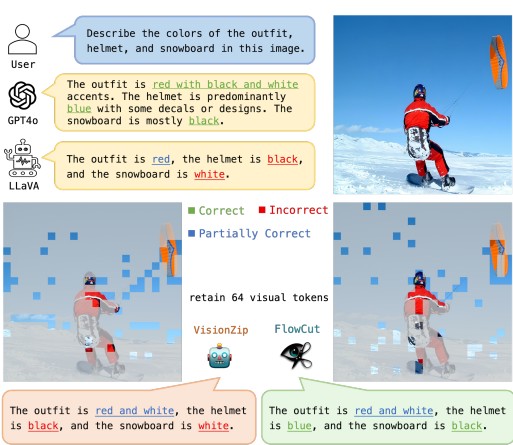

Figure 8: Example comparison of FlowCut and previous SOTA in visual question answering.

mulates diverse criteria across multiple layers and performs progressive token reduction with adaptive pruning ratios, achieving efficient inference while preserving performance. (More in appendix A.2)

## 6   Conclusion

In this work, we provide a novel perspective (information flow) for understanding the emergence of redundant tokens, revealing redundancy as a natural consequence of asymmetric attention. Through comprehensive analysis, we demonstrate the limitations of single-layer, single-criterion metrics in identifying redundancy, and propose FlowCut, an information-flow-aware pruning framework that aligns pruning decisions with the inherent dynamic information flow of tokens. Extensive experiments show that FlowCut achieves efficient inference while preserving performance.

## Acknowledgments

This work is supported by the National Natural Science Foundation of China under grants 62206102; the National Key Research and Development Program of China under grant 2024YFC3307900; the

National Natural Science Foundation of China under grants 62436003, 62376103 and 62302184; Major Science and Technology Project of Hubei Province under grant 2025BAB011 and 2024BAA008; Hubei Science and Technology Talent Service Project under grant 2024DJC078; and Ant Group through CCF-Ant Research Fund. The computation is completed in the HPC Platform of Huazhong University of Science and Technology.

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

# A  Technical Appendices and Supplementary Material

## A.1  Dataset

We evaluated our method across fifteen benchmarks: twelve for image understanding tasks and three for video understanding tasks, each assessing different aspects of multimodal intelligence.

**GQA** [20] The GQA benchmark is structured around three parts: scene graphs, questions, and images. It enriches visual content with comprehensive spatial information and object-level attributes. Questions are specifically designed to evaluate models' ability to comprehend visual scenes and reason about diverse aspects of images.

**MMBench** [42] The MMBench Benchmark evaluates models through three hierarchical levels of abilities. The first level (L-1) focuses on fundamental perception and reasoning capabilities. The second level (L-2) expands into six distinct sub-abilities, while the third level (L-3) further refines these into 20 specific dimensions. This structure enables a granular and comprehensive assessment of a model's various capabilities. Additionally, MMB-CN is the Chinese version of the benchmark.

**MME** [13] MME comprehensively evaluates models' perceptual and cognitive abilities across 14 subtasks. By employing manually constructed instruction-answer pairs and concise instructions, it effectively minimizes data leakage and ensures fair assessment of model performance.

**POPE** [35] The POPE benchmark systematically evaluates object hallucination in models through a series of binary questions about object presence in images. Using accuracy, recall, precision, and F1 score as metrics, it precisely measures hallucination levels across different sampling strategies.

**ScienceQA** [43] ScienceQA encompasses a wide array of domains, including natural, language, and social sciences, with questions hierarchically organized into 26 topics, 127 categories, and 379 skills. Through this comprehensive structure, it offers diverse scientific questions that effectively evaluate multimodal understanding, multi-step reasoning capabilities, and interpretability.

**VQA-V2** [15] VQA-V2 evaluates models' visual perception capabilities through open-ended questions about 265,016 real-world scence images. Each question contains 10 human-annotated ground truth answers, enabling thorough assessment of a model's ability to interpret and respond to visual queries.

**TextVQA** [48] TextVQA focuses on the integration of textual information within images. It evaluates a model's ability to interpret visual elements and embedded text in images through tasks, requiring both visual and textual comprehension to answer questions accurately.

**VizWiz** [18] The VizWiz benchmark contains over 31,000 visual questions created by blind individuals who photographed scenes with mobile phones while recording spoken questions. Each question includes 10 crowdsourced answers. The dataset presents distinct challenges: lower-quality images from visually impaired photographers, conversational spoken questions, and some questions that remain unanswerable due to content limitations.

**SEEDBench** [25] SEEDBench contains 19,000 human-annotated multiple-choice questions evaluating models across 12 distinct aspects. It comprehensively assesses capabilities in recognizing spatial and temporal patterns within both images and videos.

**MMVet** [60] MMVet defines six core abilities (recognition, OCR, knowledge, language generation, spatial awareness, and math) and evaluates models through 16 specific capability combinations. These integrations enable comprehensive assessment of models' performance across complex scenarios.

**LLaVA-Bench** [39] LLaVA-Bench categorizes questions into three types: conversational (simple QA), detailed description, and complex reasoning tasks, featuring 24 diverse images with 60 questions spanning indoor and outdoor scenes, memes, paintings, sketches, and more. Each image contains a manually curated detailed description and carefully selected questions, designed to assess model robustness across various prompts.

**TGIF-QA** [21] TGIF-QA extends the image question answering task to videos with 165,000 question-answer pairs based on GIFs. It introduces four task types: three requiring spatio-temporal reasoning (repetition count, repeating action, state transition) and frame QA answerable from single frames.

**MSVD-QA** [58] The MSVD-QA benchmark, based on the MSVD dataset, features 1,970 video clips with approximately 50.5K question-answer pairs. It presents open-ended questions across five

categories (what, who, how, when, where) covering diverse video content aspects, serving both video question answering and captioning tasks.

**MSRVTT-QA** [58] MSRVTT-QA comprises 10K video clips with 243K question-answer pairs, challenging models to integrate visual and temporal elements of video content. Similar to MSVD-QA, it includes five question types, evaluating models' ability to comprehend complex video content.

## A.2 Related Work

**Large Vision-Language Models** Recently, building on the success of large language models (LLMs) [2, 3, 9, 54], LVLMs [4, 8, 39, 41, 49, 53] have demonstrated impressive performance in multimodal reasoning by bridging visual and linguistic modalities. However, a critical challenge persists in processing images: the quadratic growth of visual tokens as image resolution increases, which leads to prohibitive computational costs during training and inference. High-resolution image models like LLaVA-NeXT [40] and mini-Gemini-HD [33] process thousands of tokens per image, while video models such as VideoLLaVA [36] and VideoPoet [23] demand even more tokens for multiple frames. This highlights the need for more efficient methods [28, 30, 29, 50] to extract information from visual tokens, instead of merely extending their length.

**Efficient Large Language Models** Efficient inference [11, 10, 24, 38] in LLMs is challenged by their autoregressive nature, where each token prediction depends on the full preceding context. Recently, a variety of token reduction strategies have been proposed to accelerate inference and compress the key-value (KV) cache [19]. For instance, StreamingLLM [56] retains only attention sinks and most recent tokens to shrink the KV cache, while FastGen [14] adaptively manages KV cache based on the behavior of attention head. Heavy-Hitter Oracle (H2O) [63] employs accumulated attention-based scoring to prune key-value pairs during the generation stage. These methods primarily target textual redundancy to improve inference efficiency [31, 27].

## A.3 Evaluation under Additional Settings and Comparison with More Methods

**Retain less visual tokens** To further validate our method under highly constrained token budgets, we conduct experiments with only 32 visual tokens. As shown in Table 8, our method outperforms existing methods by a notable margin under aggressive token reduction, validating its effectiveness in preserving informative content despite severe compression.

Table 8: Performance of FlowCut on LLaVA-1.5 under 32 visual token setting.

| Method | Tokens | GQA | TextVQA | VQA-V2 | POPE | SQA | VizWiz | Avg |
|---|---|---|---|---|---|---|---|---|
| LLaVA-1.5 7B (upper limit) | 576 | 61.9 | 58.2 | 78.5 | 85.9 | 69.5 | 50.0 | 100% |
| FastV | 32 | 46.2 | 51.5 | 56.0 | 35.7 | 68.3 | 49.1 | 78.7% |
| VisionZip | 32 | 51.5 | 51.7 | 65.1 | 68.3 | 68.2 | 49.9 | 88.7% |
| FlowCut (Ours) | 32 | 52.1 | 53.0 | 66.9 | 69.6 | 68.7 | 52.3 | 90.8% |

**Compare with more methods** We have already compared with several representative methods in the main text. Here, we further include more prior works for comparison: LLaVA-PruMerge [46] clusters the visual tokens via k-nearest merging algorithm and merges them. $ST^3$ [64] prunes low-importance tokens progressively across layers using token attention maps. VTW [37] removes unimportant visual tokens based on attention score at specific layers of the LLM. RCom [7] merges unimportant patch tokens based on their similarity to the CLS token and the text tokens. Faster-VLM [61] prunes vision tokens based on CLS attention scores at vision encoder's output layer.

These methods all rely on single-layer, single-metric scores (e.g., attention or similarity) to assess token importance. In contrast, our method employs multiple metrics and aggregates scores across layers using an EMD-based strategy, which helps mitigate single-metric bias and layer bias [66, 51]. We also discuss our differences with CrossGET [47] here. CrossGET injects learnable cross tokens to capture cross-modal information and merges less important tokens based on cosine similarity. In contrast, our method leverages CLS attention, performs multi-layer and multi-metric evaluation, and applies dynamic layer-wise pruning. Moreover, CrossGET requires re-training due to its learnable cross tokens while our approach is entirely training-free.

As shown in Table 9, our method better preserves performance under similar FLOPs, achieving consistently stronger results across a wide range of benchmarks. Note that for $ST^3$ , as the code is not publicly available, we follow their benchmark protocol for fair comparison.

Table 9: Comparison of FlowCut on LLaVA-1.5-7B with more methods.

| Method | TFLOPs↓ | TextVQA | VQA-V2 | POPE | MMB | SQA | | Metric | $ST^3$ [64] | FlowCut (Ours) |
|---|---|---|---|---|---|---|---|---|---|---|
| LLaVA-1.5 7B | 8.82 | 58.2 | 78.5 | 85.9 | 64.7 | 69.5 | | AI2D | 55.4 | 55.8 |
| PruMerge [46] | 1.95 | 53.5 | 65.9 | 70.7 | 56.8 | 68.5 | | SQA | 68.9 | 69.2 |
| VTW [37] (K=5) | 2.01 | 8.1 | 42.7 | 46.0 | 21.3 | 65.3 | | MMMU | 35.3 | 36.2 |
| RCom [7] | 1.96 | 55.5 | 70.4 | 72.0 | 57.9 | 69.0 | | MMB | 63.8 | 64.3 |
| FasterVLM [61] | 1.97 | 55.2 | 71.9 | 76.1 | 60.4 | 68.9 | | POPE | 85.2 | 86.1 |
| **FlowCut (Ours)** | 1.95 | 55.6 | 72.8 | 80.2 | 60.8 | 69.1 | | TFLOPs | 4.27 | 4.25 |

## A.4 More Sensitivity Analyses of Hyperparameters

**Cumulative importance mechanism's weighting coefficients**  We default to a setting of 0.5 for historical scores and 0.5 for current scores. As shown in Table 10, we further conduct a sensitivity analysis. When only the current score is used (i.e., without the cumulative strategy), the POPE performance is 83.8. The best performance is achieved at 0.5:0.5 and 0.6:0.4, while all other settings still outperform the non-cumulative baseline. This demonstrates that our method is robust to changes in the weighting coefficients and validates the effectiveness of the cumulative strategy.

Table 10: Sensitivity analysis of cumulative importance mechanism's weighting coefficients.

| history:current | 0:10 (w/o cumulative strategy ) | 1:9 | 2:8 | 3:7 | 4:6 | 5:5 | 6:4 | 7:3 | 8:2 | 9:1 |
|---|---|---|---|---|---|---|---|---|---|---|
| POPE benchmark | 83.8 (baseline) | 84.6 | 84.8 | 84.6 | 85.0 | **85.2** | **85.2** | 85.1 | 85.0 | 84.7 |

**Pruning frequency**  For pruning frequency, we default to pruning every two layers with dynamic prune ratios. As shown in Table 11: 1) Pruning every two layers yields the best performance. Performance decreases as the interval increases, and pruning every layer (n=1) performs slightly worse than n=2, which we attribute to the fact that the first pruning step under n=1 cannot leverage historical scores and must rely solely on the current layer. 2) Multi-layer pruning outperforms single-layer pruning (i.e., only pruning at the last layer), further demonstrating both robustness to this hyperparameter and the benefit of pruning redundancies as they emerge. 3) These results confirm that our method is not sensitive to the prune frequency, highlighting its robustness.

Table 11: Sensitivity analysis of pruning frequency.

| Pruning each n layer | n=1 | n=2 | n=3 | n=4 | n=6 | n=8 | n=12 | only prune at last layer |
|---|---|---|---|---|---|---|---|---|
| POPE benchmark | 84.9 | 85.2 | 84.6 | 84.6 | 84.3 | 84.3 | 84.0 | 83.9 (baseline) |

## A.5 Pseudocode

Here, we provide the pseudocode of our method. And note that method is indeed compatible with FlashAttention because it does not require the full $N \times N$ attention map. Instead, it only needs the $1 \times N$ attention value of a single CLS token or global token. When using FlashAttention, we can obtain the necessary attention values with a small, separate computation: 1) For models with a CLS token, we separately compute its $1 \times N$ attention map with the patch tokens; 2) For models without a CLS token, we derive a global token (by averaging patch tokens) and then compute its $1 \times N$ attention map. This targeted computation is highly efficient, with a complexity of just $O(n)$, adding virtually no overhead. This allows our method and FlashAttention to be used together effectively.

**Algorithm 1** Pseudocode for FlowCut

```
# Input: current layer's hidden_states and QKV
# Output: selected tokens for current layer
# QKV shape: [B, N, D] (batch size, sequence length, embedding dim)
# If the architecture without CLS token, use a globally pooled token as a substitute

if architecture_without_CLS: cls_token = output.hidden_states.mean(dim=1) # [B,D]
else: cls_token = output.hidden_states[:,0] # [B,D]

patch_token = output.hidden_states[:,1:] # [B,N-1,D]

Q_cls = Q[:, 0] # [B, D], first token is CLS token
K_patch = K[:, 1:] # [B, N-1, D], exclude CLS token

# Compute CLS-to-token attention (O(N) complexity, compatible with FlashAttention)
cls_attn = softmax(dot(Q_cls, K_patch.transpose(-1, -2)) / sqrt(D)) # [B, N-1]

# Compute semantic similarity to evaluate token-level informativeness
V_cls = V[:, 0] # [B, D]
V_patch = V[:, 1:] # [B, N-1, D]
semantic_attn = softmax(dot(V_cls, V_patch.transpose(-1, -2)) / sqrt(D)) # [B, N-1]

# Compute value norm as a proxy for token information capacity
v_score = L1_norm(V_patch) # [B, N-1]

# Fuse multiple importance criteria
importance = (normalize(cls_attn) + normalize(semantic_attn)) * v_score # [B, N-1]

# Accumulate importance across layers (EMA-like smoothing)
cumulative_score = 0.5 * cumulative_score + 0.5 * importance

# Estimate pruning ratio based on CLS attention entropy
entropy = compute_entropy(cls_attn) # [B]
entropy_ratio = entropy / log(N)
prune_ratio = (N - target_num) / sqrt(remain_layer) * (1 - entropy_ratio ** 2)

# Select top-K tokens based on cumulative importance scores
K_keep = round((1 - prune_ratio) * N)
keep_idx = topk(cumulative_score, K=K_keep) # [B, K_keep]

# Retain tokens
if not final_step: tokens = concat(cls_token, filter(patch_tokens, keep_idx))
elif final_step: tokens = filter(patch_tokens, keep_idx)
```

## A.6 Broader Impact

Our research rethink the essence of visual token reduction from a fundamental information flow perspective. Through systemical analysis of information flow patterns, we reveal that redundancy as a natural consequence of asymmetric attention. Through comprehensive analysis, we demonstrate the limitations of single-layer, single-criterion metrics in identifying redundancy, and propose FlowCut, an information-flow-aware pruning framework that aligns pruning decisions with the inherent dynamic information flow of tokens. In some cases, we observe that token pruning, by removing redundant and distracting tokens, produces more accurate answers than the non-pruned baseline. This suggests that token pruning may hold promise for mitigating hallucinations in multimodal models. Therefore, it is worth exploring in future work why token pruning helps reduce hallucinations, and how we can better leverage efficient techniques—such as token pruning and token merging—not only to accelerate inference but also to suppress hallucinations.

## A.7 More Visualization Results

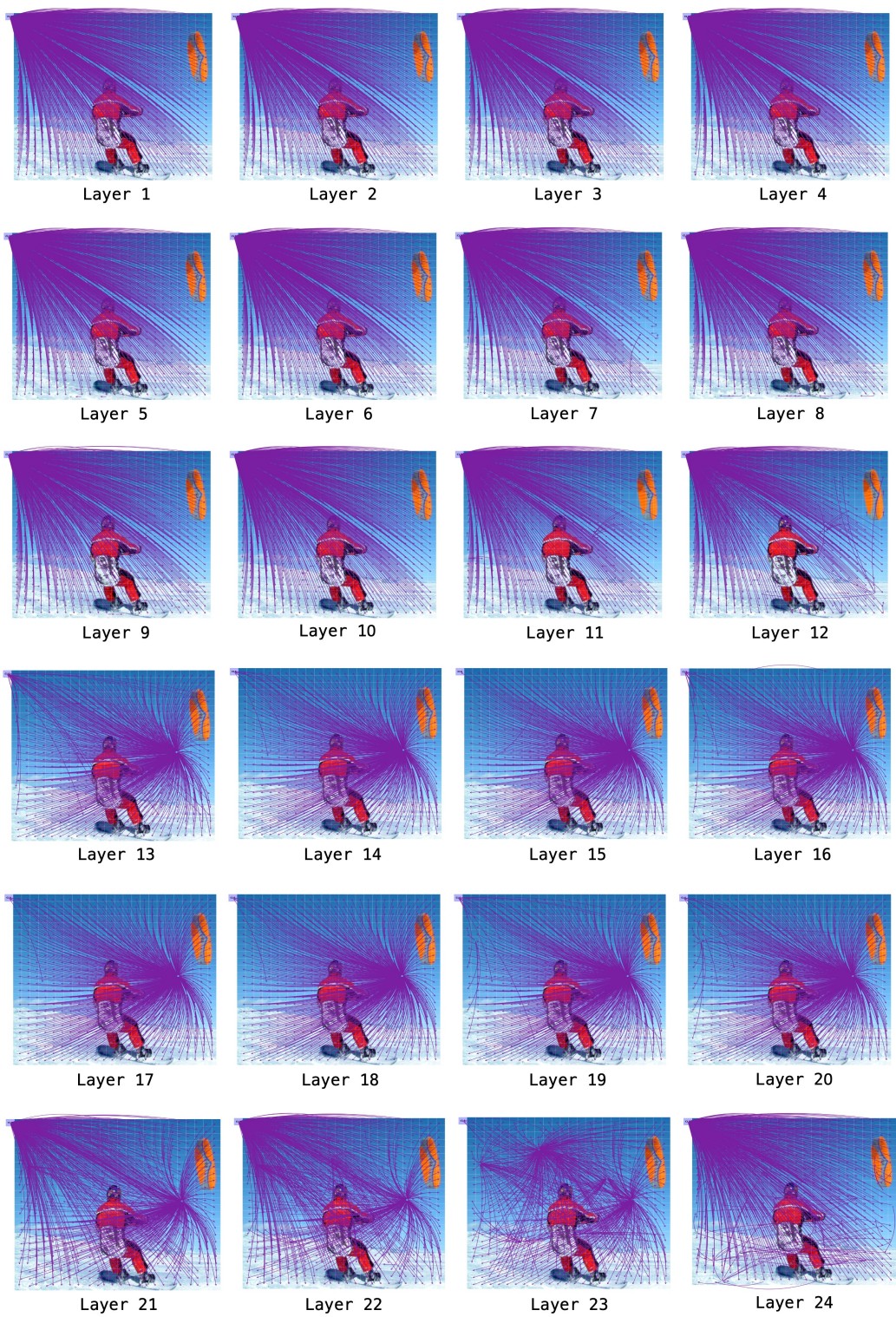

Figure 9: Visualization of information inflow main streamline.

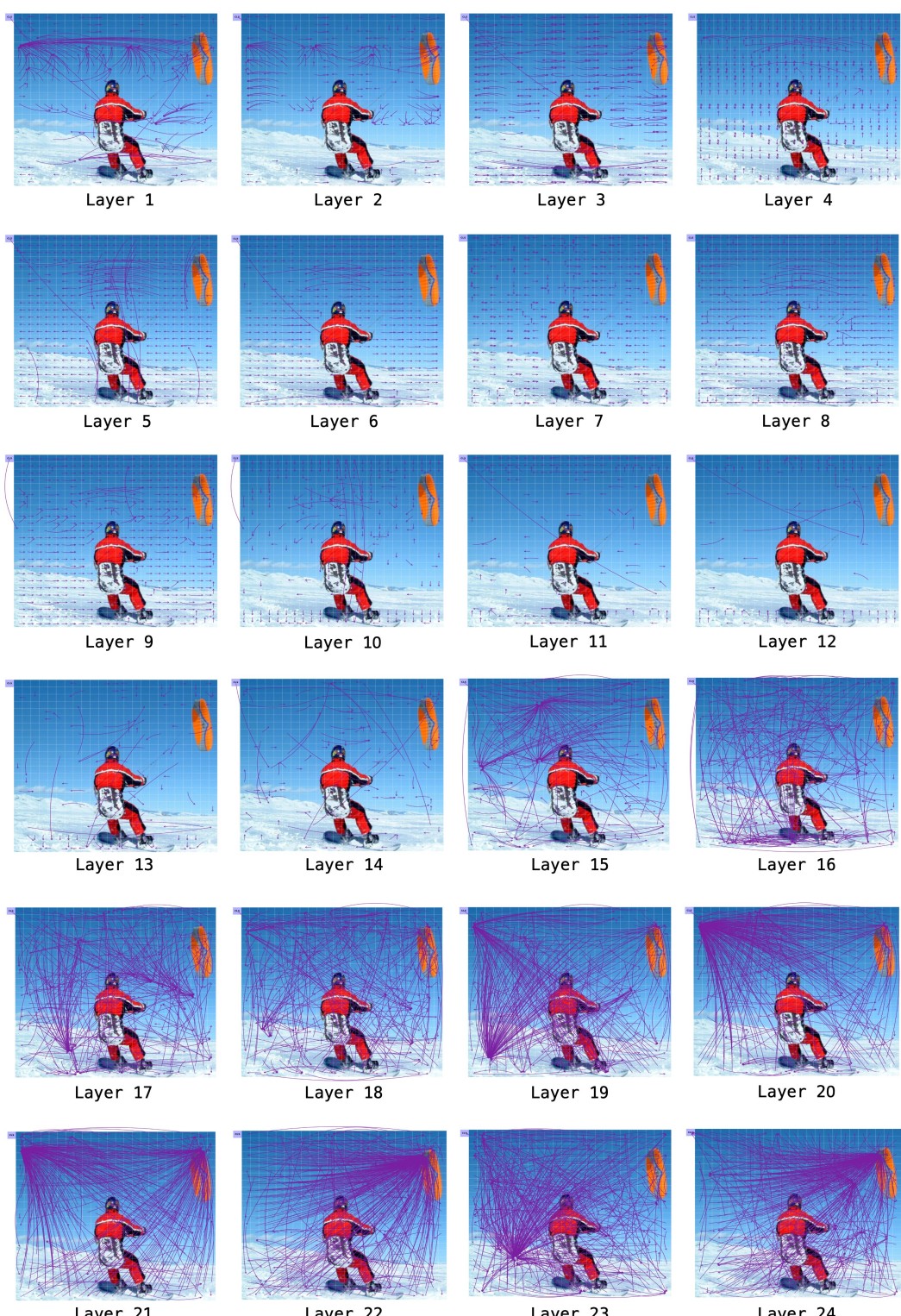

Figure 10: Visualization of information outflow main streamline.

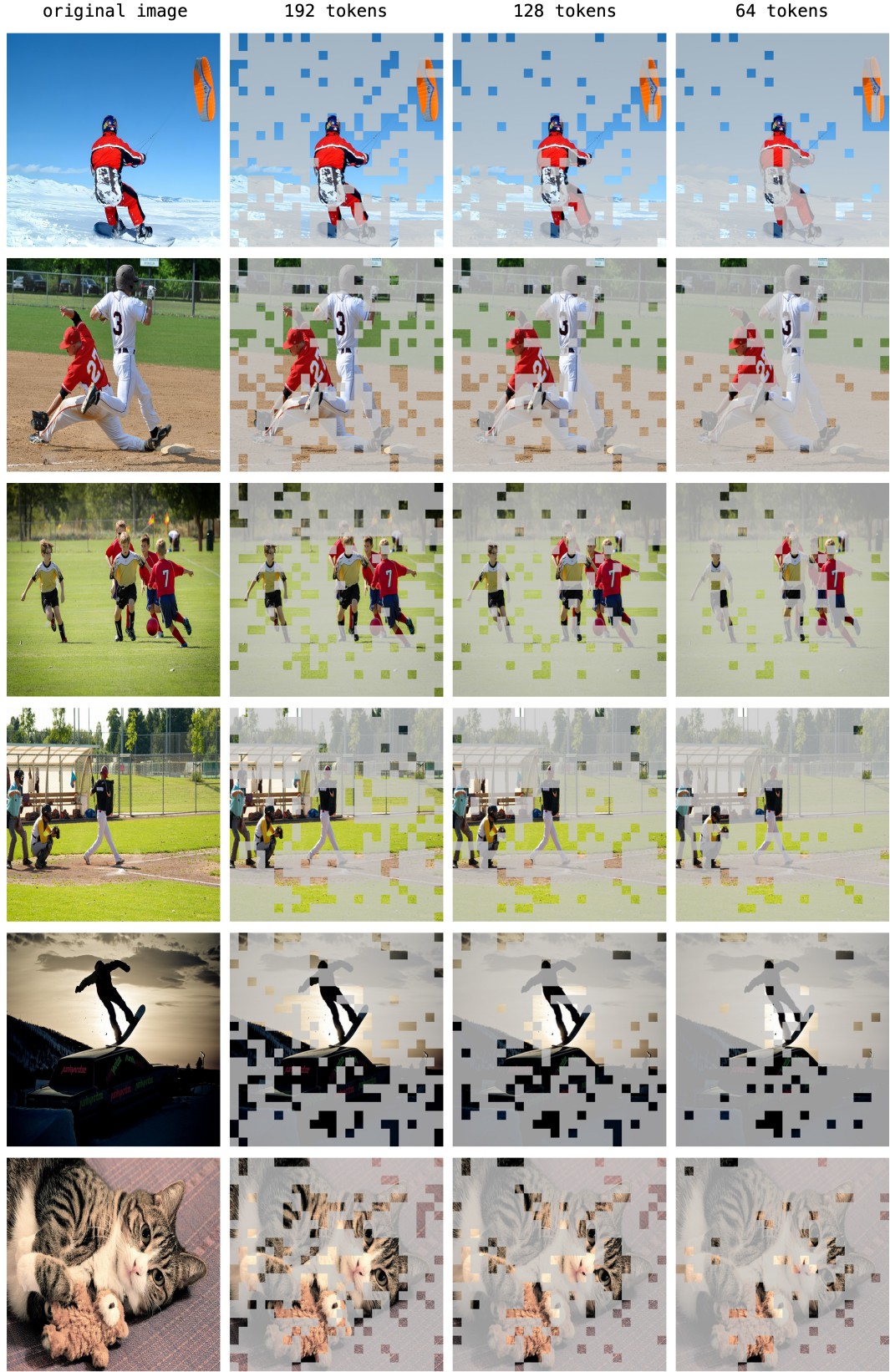

Figure 11: Sparsification Visualization examples of FlowCut on LLaVA-1.5-7B.

