# OpenReview forum: "FlowCut: Rethinking Redundancy via Information Flow for Efficient Vision-Language Models"
_NeurIPS.cc/2025/Conference — NeurIPS 2025 poster_

### Official Review · Reviewer_TJih · 2025-06-03

**Clarity:** 3
**Significance:** 3
**Originality:** 3
**Rating:** 4
**Confidence:** 5

**Summary:**

This paper provide a novel perspective (information flow) for understanding the emergence of redundant tokens, revealing redundancy as a natural consequence of asymmetric attention. Based on authors' insights,  they propose a combined criteria to measure the importance of tokens. The method is tested on multiple benchmarks, demonstrating robustness.

**Questions:**

1.What is the overhead of the proposed work？And how do authors reproduce the prior works in Tab. 5, using flash attention or not?

2. In Eq. 6, the authors combine historical and current criteria by a equal weight 0.5, what about others weights? Moreover, the token importance in last layer is used in the current layer, does it make sense?

3. Token pruning is implemented in every two layers，why？How about three or more layers?

4.What about the performance on image caption task like coco caption, nocaps and flickr30k?

**Ethical Concerns:**

["NO or VERY MINOR ethics concerns only"]

**Final Justification:**

Thanks for authors' rebuttal,  I have raised my score. I suggest the authors include the new experiments and discussion in their revision.

**Limitations:**

The authors argue that they discuss the limitations of the work in the conclusion, but in fact not. For limitations, please refer to weaknesses and questions.

**Quality:**

3

**Strengths And Weaknesses:**

Strengths

1. The figures are visually good.
2. The experimental evaluations are conducted on several MLLMs and video benchmarks, the results are promising.
3. The method is clear and easy to follow.

Weaknesses

1. The author proposed a new concept called information flow, but it is essentially just the attention in the model. This concept actually diminishes the readability and comprehensibility of the paper.
2. The insights are neither novel nor appealing, as similar observations have been made in many previous articles on Vision Transformers.
3. The proposed method is just attention-based and not aligned well with insights, what is the relationship between it and information flow?
4. Some benchmarks are missed, see questions.
5. Literature review is not comprehensive enough, some prior works should be discussed in related work or compared in experiments:

[1] LLaVA-PruMerge: Adaptive Token Reduction for Efficient Large Multimodal Models. Arxiv24

[2] ST3: Accelerating Multimodal Large Language Model by Spatial-Temporal Visual Token Trimming. AAAI25

[3] [CLS] Attention is All You Need for Training-Free Visual Token Pruning: Make VLM Inference Faster. Arxiv24

[4] Recoverable Compression: A Multimodal Vision Token Recovery Mechanism Guided by Text Information. AAAI25

[5] Boosting Multimodal Large Language Models with Visual Tokens Withdrawal for Rapid Inference. AAAI25

---

> ### Author Rebuttal · Authors · 2025-07-29
>
> ### 1. Clarify misunderstanding of our work's insight and method (for weakness 1,2,3)
> There may be some misunderstandings about our work. We introduces a unified, bottom-up perspective (*information flow*) by modeling token-to-token interaction to explore the propagation pattern of visual information and understand the emergence of redundant tokens. **Attention serves as only one component,** our perspective on information flow extends beyond attention: we analyze and visualize the evolution of information flow across layers, quantify each token's information content by Value-vector magnitudes, and assess whether tokens contain semantically meaningful information by semantic similarity. All these analyses are grounded in our proposed *information flow* view. As Reviewer T8rC noted: *“providing a certain theoretical foundation and offering potential for further improvement”*, and Reviewer NDVm said: *“The Insights module rigorously analyzes visual token redundancy from a dynamic information-flow perspective”*.
>
> **For the insight:** Our proposed *information flow* goes beyond merely referring to the attention in the model. 1) We model the token interaction process from an information flow perspective and primarily analyze the patterns and trends of information interaction, with some analyses illustrated through attention map visualizations rather than focusing on attention itself. 2) We analyze and visualize the evolution of information flow across layers, showing that lower layers exhibit local, neighborhood-level interactions, while higher layers engage in long-range semantic interactions. 3) We incorporate Value-vector magnitudes to quantify the amount of information each token carries and semantic similarity to assess whether tokens contain semantically redundant or meaningful information.
>
> **For the method:** Each module of our method is motivated by phenomena uncovered through our information flow analysis: 1) Based on the observed layer-wise differences in information interaction patterns, we propose an EMD-like cumulative scoring strategy that iteratively updates token importance by aggregating both current and historical interaction evidence; 2) Based on the finding that redundancy emerges progressively as attention concentrates over depth, driven by increasingly focused semantic interactions, we introduce an adaptive mechanism to determine the pruning ratio for each layer accordingly; 3) Based on the analysis that attention alone does not fully capture token contribution, we additionally incorporate the Value-vector L1 norm and semantic similarity to estimate each token’s contribution to information gain.
>
> ### 2. Experiments on more benchmarks (for weakness 4 & question 4)
>
> In the main text, we have already evaluated our method on 12 image benchmarks and 3 video benchmarks. Here, we further include results on three widely-used image captioning benchmarks: COCO 2017 Caption, NoCaps, and Flickr30k. We evaluate our method under both 192 and 128 token settings. The results show that our method achieves strong performance on these benchmarks, offering substantial inference acceleration with minimal performance degradation.
>
> |                     |Token|COCO|Nocaps|Flickr30k|
> | ------------------ | :---: | :------: | :------: | :------: |
> | LLaVA-1.5 7B (upper limit) |576|110.5|105.5|75.2|
> | FastV                      |192|109.1|104.2|73.5|
> | Ours                       |192|**110.7**|**105.3**|**74.9**|
> | FastV                      |128|105.9|101.2|71.0|
> | Ours                       |128|**107.8**|**102.4**|**73.6**|
>
> ### 3. Compare with more works (for weakness 5)
>
> We have already compared with several representative methods in the main text. Here, we further include more prior works for comparison: PruMerge clusters the visual tokens via k-nearest neighbormerges algorithm and merge them. ST$^3$ prunes low-importance tokens progressively across layers using token attention maps. VTW removes unimportant visual tokens based on attention score at specific layers of the LLM. RCom merges unimportant patch tokens based on their similarity to the CLS token and the text tokens. FasterVLM prunes vision tokens based on CLS attention scores at vision encoder's output layer.
>
> These methods all rely on single-layer, single-metric scores (e.g., attention or similarity) to assess token importance. In contrast, our method employs multiple metrics and aggregates scores across layers using an EMD-based strategy, which helps mitigate single-metric bias and layer bias.
>
> As shown in the tables below, our method better preserves performance under similar FLOPs, achieving consistently stronger results across a wide range of benchmarks. Note that for ST$^3$, as the code is not publicly available, we follow their benchmark protocol for fair comparison.
>
> | Methods            | TFLOPs $\downarrow$ | TextVQA  |  VQA-V2  |   POPE   |   MMB    |   SQA    |
> | ------------------ | :------------: | :------: | :------: | :------: | :------: | :------: |
> | Llava-1.5 7B       |8.82|58.2|78.5|85.9|64.7|69.5|
> | PruMerge [1]       |1.95|53.5|65.9|70.7|56.8|68.5|
> | VTW [5] (K=5)      |2.01|8.1|42.7|46.0|21.3|65.3|
> | RCom [4]           |1.96|55.5|70.4|72.0|57.9|69.0|
> | FasterVLM [3]      |1.97|55.2|71.9|76.1|60.4|68.9|
> | **FlowCut (Ours)** |1.95|**55.6**|**72.8**|**80.2**|**60.8**|**69.1**|
>
> |                    | TFLOPs $\downarrow$ |   AI2D   |   SQA    |   MMMU   |   MMB    |   POPE   |
> | ------------------ | :-----------------: | :------: | :------: | :------: | :------: | :------: |
> | ST$^3$ [2]         |  4.27  |   55.4   |   68.9   |   35.3   |   63.8   |   85.2   |
> | **FlowCut (Ours)** |  4.25  | **55.8** | **69.2** | **36.2** | **64.3** | **86.1** |
>
> [1] LLaVA-PruMerge: Adaptive Token Reduction for Efficient Large Multimodal Models. Arxiv24
>
> [2] ST3: Accelerating Multimodal Large Language Model by Spatial-Temporal Visual Token Trimming. AAAI25
>
> [3] [CLS] Attention is All You Need for Training-Free Visual Token Pruning: Make VLM Inference Faster. Arxiv24
>
> [4] Recoverable Compression: A Multimodal Vision Token Recovery Mechanism Guided by Text Information. AAAI25
>
> [5] Boosting Multimodal Large Language Models with Visual Tokens Withdrawal for Rapid Inference. AAAI25
>
> ### 4. Overhead of our work (for question 1)
>
> We have reported the overhead of our method in Table 5 of the main text. The FLOPs reported for both our method and other baselines in Table 5 are measured without using Flash Attention. This is because prior works (e.g., SparseVLM**, **FastV, and VisionZip) are not compatible with Flash Attention. Therefore, to ensure a fair comparison, we report all FLOPs under the same setting (without Flash Attention).
>
> It is worth noting that our method is compatible with Flash Attention, and using them together can further accelerate inference. We report this combination in Appendix Table 7, where we observe a significant speed-up in training time when both are used.
>
> ### 5. More explanation of cumulative importance strategy (for question 2)
> As shown in main text Figure 2 and Figure 4, the information contribution of the same token varies across layers. Therefore, determining token importance based solely on the current layer introduces layer bias, as a token might appear unimportant in the current layer but be crucial in others. To address this, we adopt an EMD-like cumulative scoring strategy, where token importance is updated as: cumulative = 0.5 × current + 0.5 × previous cumulative. This approach effectively integrates both local and historical importance signals. Its effectiveness is supported by the ablation study in Table 6 of the main text, as well as the sensitivity analysis discussed in our Answer 6.
>
> ### 6. More sensitivity analyses of hyperparameters (for question 2, 3)
> For the cumulative importance evaluation mechanism's weighting coefficients, we default to a setting of 0.5 for historical scores and 0.5 for current scores. We further conduct a sensitivity analysis, with three key observations: 1) When only the current score is used (i.e., without the cumulative strategy), the POPE performance is 83.8. Incorporating historical scores consistently improves performance across all coefficient settings, validating the effectiveness of our cumulative strategy. 2) The best performance is achieved at 0.5:0.5 and 0.6:0.4, while all other settings still outperform the non-cumulative baseline, indicating robustness to coefficient changes. 3) These results confirm that our method is not sensitive to the weighting coefficients, demonstrating its robustness.
>
> | history : current | 0:10 (w/o cumulative strategy ) | 1:9  | 2:8  | 3:7  | 4:6  |   5:5    |   6:4    | 7:3  | 8:2  | 9:1  |
> | :---------------: | :-----------: | :--: | :--: | :--: | :--: | :------: | :------: | :--: | :--: | :--: |
> |  POPE benchmark   |83.8 (baseline)| 84.6 | 84.8 | 84.6 | 85.0 | **85.2** | **85.2** | 85.1 | 85.0 | 84.7 |
>
> For pruning frequency, we default to pruning every two layers with dynamic prune ratios. Additional sensitivity experiments show: 1) Pruning every two layers yields the best performance. Performance decreases as the interval increases, and pruning every layer (n=1) performs slightly worse than n=2. We attribute this to the fact that the first pruning step under n=1 cannot leverage historical scores and must rely solely on the current layer. 2) Multi-layer pruning outperforms single-layer pruning (i.e., only pruning at the last layer), again demonstrating both robustness to this hyperparameter and the benefit of pruning redundancies as they emerge.  3) These results confirm that our method is not sensitive to the prune frequency, demonstrating its robustness.
>
> | Pruning each n layer | n=1  |   n=2    | n=3  | n=4  | n=6  | n=8  | n=12 | only prune at last layer |
> | :------------------: | :--: | :------: | :--: | :--: | :--: | :--: | :--: | :-----------: |
> |    POPE benchmark    | 84.9 | **85.2** | 84.6 | 84.6 | 84.3 | 84.3 | 84.0 |83.9 (baseline)|

---

> ### Comment · Reviewer_TJih · 2025-08-05
>
> Thanks for authors' rebuttal,  I have raised my score. I suggest the authors include the new experiments and discussion in their revision.

---

> > ### Author Response · Authors · 2025-08-05
> >
> > We sincerely thank you for acknowledging our response. We will carefully incorporate the new experiments and discussions into the final version

---

### Official Review · Reviewer_NDVm · 2025-06-27

**Clarity:** 4
**Significance:** 4
**Originality:** 3
**Rating:** 4
**Confidence:** 4

**Summary:**

This paper targets the inference inefficiency of large-scale vision–language models (LVLMs) caused by redundant visual tokens by proposing an information-flow-based pruning method. The authors dynamically adjust pruning strength at each transformer layer according to the CLS token’s attention entropy, score tokens using attention weight, semantic similarity to CLS, and Value-vector norm, and aggregate importance across layers to avoid the instability of single-layer criteria. On multiple LVLM architectures and 12 image + 3 video benchmarks, FlowCut dramatically reduces token count—up to 94.4% fewer tokens—while preserving 91.9% inference accuracy. Under fixed final-token budgets, its layer-wise adaptive pruning achieves the greatest cumulative computation reduction compared to prior methods.

**Questions:**

1. The multi-criteria evaluation and cumulative flow tracking rely on several hyperparameters (e.g., weighting coefficients, pruning frequency). Could the authors include ablation studies or sensitivity analyses to demonstrate robustness across a reasonable range of these parameters?

2. FlowCut currently uses attention entropy and the Value-vector L₁ norm to characterize information flow. How can it avoid mistakenly pruning critical tokens in dense multi-object or heavily occluded scenes?

3. Some readers may not be familiar with Vision Transformer details; terms like "CLS" should be defined at first mention or briefly explained to improve clarity.

4. Long video inputs dramatically increase token counts and introduce large inter-frame semantic shifts. Can FlowCut's layer-wise adaptive pruning maintain high-quality inference under these conditions?

**Ethical Concerns:**

["NO or VERY MINOR ethics concerns only"]

**Limitations:**

Yes

**Quality:**

3

**Strengths And Weaknesses:**

Pros

1. The framework is hierarchically structured and methodologically novel, achieving significant speed–accuracy gains with an intuitive combination of strategies.

2. The Insights module rigorously analyzes visual token redundancy from a dynamic information-flow perspective, presenting core observations step by step with clear logic.

3. Experiments are comprehensive across multiple LVLM architectures and 12 image + 3 video benchmarks, using diverse metrics to fully demonstrate the efficiency–effectiveness trade-off.

Cons

1. The multi-criteria evaluation and cumulative flow tracking involve several hyperparameters (e.g., weighting coefficients) without ablation or sensitivity analyses to validate robustness.

2. Evaluation is limited to the LLaVA family and Qwen2-VL; applicability to other vision encoders or CLS-free architectures remains unverified.

3. Reliance solely on attention entropy and Value-vector L₁ norm to characterize information flow may misjudge redundancy in complex scenes (e.g., dense multi-object overlaps).

---

> ### Author Rebuttal · Authors · 2025-07-29
>
> **Thank for your valuable input and appreciation on our work**
>
> ### 1. More sensitivity analyses of hyperparameters (for weakness 1 & question1)
>
> For the cumulative importance evaluation mechanism's weighting coefficients, we default to a setting of 0.5 for historical scores and 0.5 for current scores. We further conduct a sensitivity analysis, with three key observations: 1) When only the current score is used (i.e., without the cumulative strategy), the POPE performance is 83.8. Incorporating historical scores consistently improves performance across all coefficient settings, validating the effectiveness of our cumulative strategy. 2) The best performance is achieved at 0.5:0.5 and 0.6:0.4, while all other settings still outperform the non-cumulative baseline, indicating robustness to coefficient changes. 3) These results confirm that our method is not sensitive to the weighting coefficients, demonstrating its robustness.
>
> | history : current | 0:10 (w/o cumulative strategy ) | 1:9  | 2:8  | 3:7  | 4:6  |   5:5    |   6:4    | 7:3  | 8:2  | 9:1  |
> | :---------------: | :-----------------------------: | :--: | :--: | :--: | :--: | :------: | :------: | :--: | :--: | :--: |
> |  POPE benchmark   |         83.8 (baseline)         | 84.6 | 84.8 | 84.6 | 85.0 | **85.2** | **85.2** | 85.1 | 85.0 | 84.7 |
>
> For pruning frequency, we default to pruning every two layers with dynamic prune ratios. Additional sensitivity experiments show: 1) Pruning every two layers yields the best performance. Performance decreases as the interval increases, and pruning every layer (n=1) performs slightly worse than n=2. We attribute this to the fact that the first pruning step under n=1 cannot leverage historical scores and must rely solely on the current layer. 2) Multi-layer pruning outperforms single-layer pruning (i.e., only pruning at the last layer), again demonstrating both robustness to this hyperparameter and the benefit of pruning redundancies as they emerge.  3) These results confirm that our method is not sensitive to the prune frequency, demonstrating its robustness.
>
> | Pruning each n layer | n=1  |   n=2    | n=3  | n=4  | n=6  | n=8  | n=12 | only prune at last layer |
> | :------------------: | :--: | :------: | :--: | :--: | :--: | :--: | :--: | :----------------------: |
> |    POPE benchmark    | 84.9 | **85.2** | 84.6 | 84.6 | 84.3 | 84.3 | 84.0 |     83.9 (baseline)      |
>
> ### 2. Our method can apply on cls-free architecture (for weakness 2)
>
> Our method is fully applicable to CLS-free architectures. We would like to clarify that Qwen2-VL is a CLS-free architecure, and as shown in main text Table 3 our method demonstrates strong performance on this architecture as well (we copy result here for convenient lookup). This confirms that the effectiveness of our approach is not limited to CLS-based models.
>
> | Method                    |         Tokens          |   GQA    |   MMB    | MMB$^{CN}$ |   MME    |   POPE   |   SQA    | TextVQA  |   Avg.    |
> | ------------------------- | :---------------------: | :------: | :------: | :--------: | :------: | :------: | :------: | :------: | :-------: |
> | Qwen2-VL 7B (upper limit) | 100%|   61.9   |   79.9   |    79.5    |   2338   |   87.2   |   85.1   |   82.2   |   100%    |
> | FastV                     |          33.3%          |   58.0   |   76.1   |    73.9    |   2130   |   82.1   |   80.0   |   77.3   |   93.6%   |
> | Ours                      |          33.3%          | **60.5** | **79.2** |  **78.2**  | **2335** | **86.0** | **84.0** | **81.1** | **98.7%** |
> | FastV                     |          22.2%          |   56.7   |   74.1   |    72.4    |   2031   |   79.2   |   78.3   |   72.0   |   90.4%   |
> | Ours                      |          22.2%          | **59.2** | **77.8** |  **76.9**  | **2310** | **84.6** | **80.5** | **78.3** | **96.5%** |
> | FastV                     |          11.1%          |   51.9   |   70.1   |    63.8    |   1962   |   76.1   |   75.8   |   60.3   |   83.6%   |
> | Ours                      |          11.1%          | **56.4** | **72.6** |  **72.5**  | **2252** | **81.8** | **78.2** | **68.9** | **91.3%** |
>
>
>
> ### 3. More complex scenes  (for weakness3 & question 2)
>
> Our method not only utilizes attention-based signals (attention entropy and attention score) and the Value-vector L₁ norm to characterize information flow, but also introduces a semantic effectiveness metric: $\mathrm{semantic\_similarity} = \text{softmax}\left( \frac{V_{\text{cls}} \cdot V_{\text{patch}}^\top}{\sqrt{D}} \right)$, which evaluates whether a token contains semantically meaningful content. Compared to prior work relying solely on attention scores, our multi-criteria approach ensures that preserved tokens are both highly attended and semantically informative.
>
> For more complex scenes, we would like to point out that VQA-v2 benchmark includes images with dense multi-object scenes, where our method performs strongly (main text Table 1).
>
> Here, we evaluate on the more challenging Referring Expression Comprehension (REC) task from the RefCOCO benchmark, which requires the model to understand textual instructions and detect target objects in images that often contain dense multi-object and heavily occluded scenes. For example, given an image containing multiple people in different outfits and the instruction “woman with green shirt,” the model must correctly locate the intended target. Our method performs well under such conditions, demonstrating its ability to prune redundant tokens while preserving those critical for fine-grained understanding.
>
> | RefCoCo                     |               Tokens               | val       | testA     | testB     |
> | --------------------------- | :--------------------------------: | --------- | --------- | --------- |
> | Qwen2.5-VL-7B (upper limit) | 100%  | 89.45     | 92.56     | 85.16     |
> | FastV                       |             retain 50%             | 73.85     | 73.38     | 74.21     |
> | Ours                        |             retain 50%             | **87.29** | **90.93** | **83.06** |
>
> ### 4. Long video scenes (for question 4)
>
> As shown in Table 4 of the main text, we have already evaluated our method on three video understanding benchmarks, demonstrating that FlowCut can effectively accelerate inference with negligible performance drop in video scenarios. To further assess its robustness on long-form inputs, we additionally evaluate on EgoSchema, a long-form video question answering dataset. The results show that FlowCut continues to perform well under long video settings:
>
> |                              | Tokens | EgoSchema |
> | ---------------------------- | :----: | :-------: |
> | Video-LLaVA 7B (upper limit) |  2048  |   38.4    |
> | FastV                        |  256   |   33.7    |
> | Ours                         |  256   | **37.9**  |
>
> ### 5.  Additional explanation of the CLS token (for question 3)
>
> Thank you very much for your suggestion. We will add an additional explanation of the CLS token to help readers unfamiliar with Vision Transformers better understand it:
>
> > The CLS (classification) token is a special learnable token prepended to the input sequence in Vision Transformers, whose final representation is typically used for classification or as a summary of the image.

---

> > ### Comment · Reviewer_NDVm · 2025-08-05
> >
> > Thank you for the detailed response. I will maintain my original score.

---

> > > ### Author Response · Authors · 2025-08-05
> > >
> > > We sincerely thank you for engaging with our response and recognizing our work.

---

### Official Review · Reviewer_T8rC · 2025-07-02

**Clarity:** 3
**Significance:** 3
**Originality:** 3
**Rating:** 5
**Confidence:** 4

**Summary:**

This paper first analyzes the information propagation in Transformers from shallow to deep layers using attention scores. It finds that the CLS token functions as a relay, helping most patch tokens gradually expand their attention from neighboring positions in shallow layers to longer-range dependencies in deeper layers, eventually imitating the CLS token by focusing on a few specific tokens. This insight inspires a dynamic pruning strategy. By integrating attention scores, similarity to the CLS token, and the L1 norm, the authors propose a layer-wise importance metric, which is further accumulated across layers. This leads to an efficient and powerful dynamic pruning method that outperforms traditional static pruning approaches.

**Questions:**

1. During forward propagation, is dynamic pruning performed simultaneously at each layer? If so, does it mean that pruning decisions made in earlier layers will influence the pruning results in later layers?
2. Additionally, if that's the case, since the number of pruned tokens per layer is related to attention entropy, how is it ensured that the final number of tokens matches the target? Is the final pruning step forcibly adjusted to meet the target count?
3. In Section 2 of the main paper, the analysis of information propagation is based on a tokenizer that includes a CLS token. If a tokenizer without a CLS token is used, and the result of global pooling is treated as the CLS token, would similar conclusions still hold?
4. What is the design principle behind the combination of multiple metrics? Why are both addition and multiplication used?
5. In Equation (4) of the main paper, what does $I^r$ represent? It is not mentioned earlier in the text—should it instead be $I^d$?

**Ethical Concerns:**

["NO or VERY MINOR ethics concerns only"]

**Final Justification:**

The rebuttal clarified my main questions. I have no remaining concerns and will maintain my original score.

**Limitations:**

Yes.

**Paper Formatting Concerns:**

None.

**Quality:**

4

**Strengths And Weaknesses:**

**Strengths**
1. The proposed method not only achieves faster computation but also delivers better performance.
2. It is motivated by the perspective of information propagation, providing a certain theoretical foundation and offering potential for further improvement.
3. The idea of dynamic pruning offers a more flexible and elegant solution.

**Weaknesses**
1. The paper does not provide a detailed rationale behind the design choices for combining multiple metrics.

---

> ### Author Rebuttal · Authors · 2025-07-29
>
> **Thank for your valuable input and appreciation on our work**
>
> ### 1. More detailed rationale behind the design for multiple metric (for weakness 1 & question 4)
>
> Most existing methods adopt a single metric, typically the attention score, as the sole criterion for token importance:
> $\mathrm{cls\_attn} = \text{softmax}\left( \frac{Q_{\text{cls}} \cdot K_{\text{patch}}^\top}{\sqrt{D}} \right).$
> However, we argue that this is insufficient. From an information flow perspective, attention alone does not reliably indicate how much information a token conveys. According to the attention computation formula:
>
> $\text{Attn} = \text{softmax}(QK^\top/\sqrt D), \text{output}=Attn\cdot V$
>
> even if a token receives a high attention weight, it does not necessarily contribute significant information to the final representation. Its actual contribution also depends on the magnitude of its value vector $V$. If the attention score is high but $V$ is close to zero, the token’s contribution is negligible (as shown in main text Figure 4, where some tokens exhibit high attention but nearly zero $V$). Therefore, we introduce the $l_1\text{-}norm$ of $V$ as an additional metric, capturing the actual information carried by the token.
>
> However, some tokens may exhibit large value magnitudes while lacking semantically useful information (e.g., register tokens), making it unreliable to assess their true contribution based solely on attention and value magnitude. To this end, we introduce a semantic similarity-based metric to assess whether a token carries meaningful information: $\mathrm{semantic\_attn} = \text{softmax}\left( \frac{V_{\text{cls}} \cdot V_{\text{patch}}^\top}{\sqrt{D}} \right),$ which measures the alignment between the CLS token and each patch token in the value space. Since the CLS token aggregates semantically important content, a high similarity indicates that the token likely contributes useful information.
>
> By aligning with the structure of the attention output computation: $\text{softmax}(QK^\top / \sqrt D) \cdot V$, we design our combined metric as: $ ( \mathrm{semantic\_attn} + \mathrm{cls\_attn}) \times \|V\|_1.$ This formulation ensures that the preserved tokens are both highly attended and semantically informative, with non-trivial content. That's why we use both addition and multiplication. As demonstrated in the ablation studies in the main text, this multi-metric design consistently outperforms single-metric baselines, confirming its effectiveness.
>
> ### 2. More details about adaptive pruning strategy (for question 1, 2)
>
> For the adaptive pruning strategy, we calculate the number of tokens to prune and perform pruning every two layers. This design ensures that each pruning step considers both current and historical token importance, avoiding layer bias caused by inconsistent token relevance across layers.
>
> Moreover, pruning decisions in later layers are influenced by earlier ones. Since redundancy emerges progressively as attention becomes more concentrated, pruning tokens as soon as they become redundant is more effective than deferring pruning to the final layer. The ablation study in the main text (Table 6) confirms this. Early removal prevents redundant tokens from continuously consuming attention, allowing more information to flow toward useful tokens.
>
> In addition, the number of tokens pruned in each layer depends on how many have already been removed. As shown in Equation (3) in the main text, we calculate the pruning ratio based on the difference between the target number of tokens and the cumulative number already pruned.
>
> A final pruning step forcibly removes the remaining tokens to exactly meet the target count.
>
> ### 3. Conclusions still hold in non-CLS architecture (for question 3)
>
> Our conclusions remain valid for non-CLS architectures. To support this, we additionally report the attention entropy of the CLS token in *LLaVA-1.5 (with CLS)* and the global token in *LLaVA-1.5 (without CLS)*, where the global pooled token is treated as the CLS equivalently, on the same input sample. While the absolute values differ slightly, **the overall trends across layers are highly consistent**, indicating that the global token plays a similar role in capturing information flow.
>
> | Layer                     | 2    | 4    | 6    | 8    | 10   | 12   | 14   | 16   | 18   | 20   | 22   |
> | ------------------------- | ---- | ---- | ---- | ---- | ---- | ---- | ---- | ---- | ---- | ---- | ---- |
> | CLS token attn entropy    | 6.2  | 6.1  | 5.9  | 5.8  | 5.3  | 3.8  | 4.4  | 3.8  | 3.9  | 4.2  | 4.3  |
> | global token attn entropy | 6.2  | 6.0  | 5.9  | 5.9  | 5.5  | 3.7  | 4.2  | 3.6  | 3.6  | 4.1  | 4.2  |
>
> Furthermore, as shown in Table 3 of the main text, we apply our method to Qwen2-VL, whose vision encoder is CLS-free. The model still achieves strong performance, demonstrating the effectiveness of our approach in CLS-free architectures.
>
> ### 4. Minor Writing error (for question 5)
> Thank you very much for your careful correction. Equation 4 should indeed be $I^d$ instead of $I^r$. We will take extra care to correct this in the final version to prevent any minor writing errors.

---

> > ### Comment · Reviewer_T8rC · 2025-08-02
> >
> > Thank you for the detailed response. My concerns have been fully addressed. I will maintain my original score.

---

> > > ### Author Response · Authors · 2025-08-03
> > >
> > > We sincerely thank you for engaging with our response and recognizing our work.

---

### Official Review · Reviewer_4FwU · 2025-07-03

**Clarity:** 2
**Significance:** 2
**Originality:** 2
**Rating:** 4
**Confidence:** 3

**Summary:**

This paper introduces FlowCut, which is a token pruning method for reducing visual tokens in multimodal LLMs. While previous methods typically use attention scores from a single layer to identify redundant tokens, FlowCut proposes an information flow approach that identifies redundant tokens progressively through attention across multiple layers. Specifically, FlowCut suggests a layer-wise adaptive pruning ratio, a multi-criteria token scoring system, and a cumulative importance evaluation mechanism to aggregate importance over layers. Experiments are conducted on widely used multimodal LLMs such as LLaVA-1.5 and LLaVA-NeXT, showing that FlowCut outperforms some previous baselines.

**Questions:**

See Weaknesses please

**Ethical Concerns:**

["NO or VERY MINOR ethics concerns only"]

**Final Justification:**

Thank the authors for providing a rebuttal. My concerns have been addressed, and I have raised my score. I'd suggest the authors carefully revise and include the new experiments and discussion to their final version.

**Limitations:**

Yes

**Quality:**

3

**Strengths And Weaknesses:**

### Strengths
* This paper is well-organized
* Extensive visualization is provided

### Weaknesses
* It seems that this method is incompatible with FlashAttention since it depends on the attention value.

* For better clarity, I'd suggest the authors include pseudocode or an algorithm to describe their method precisely.

* It would be better to have experimental results under a very small token number, as there is too much redundancy in the visual domain.

* This paper shares some similar designs, such as the utilization of the CLS token with a prior work Crossget [1]. I’d suggest the authors provide a comparison or discussion.

[1] CrossGET: Cross-Guided Ensemble of Tokens for Accelerating Vision-Language Transformers.  (ICML 2024).

---

> ### Author Rebuttal · Authors · 2025-07-26
>
> **Thank for your valuable input.**
>
> ### 1. Our method is indeed compatible with FlashAttention (for weakness 1)
>
> Our method is indeed compatible with FlashAttention because it does not require the full N×N attention map. Instead, it only needs the 1×N attention value of a single cls token or global token. When using FlashAttention, we can obtain the necessary attention values with a small, separate computation: 1) For models with a `CLS` token, we separately compute its 1×N attention map with the patch tokens; 2) For models without a `CLS` token, we derive a global token (by averaging patch tokens) and then compute its 1×N attention map. This targeted computation is highly efficient, with a complexity of just O(n), adding virtually no overhead. This allows our method and FlashAttention to be used together effectively.
>
> The training process for our results in Appendix Table 7 utilized both our method (FlowCut) and FlashAttention, achieving a significant speedup. We copy the result here for convenient lookup.
>
> | Method (using flash attention) | Token | Training Time (w/ flash attention) |
> | ------------------------------ | :---: | :--------------------------------: |
> | LLaVA-1.5 7B (upper limit)     |  576  |               12.97h               |
> | Ours                           |  128  |               8.02h                |
>
> ### 2. Pseudocode (for weakness 2)
>
> Thank you for the valuable suggestion. Due to markdown limitations, we provide a simplified pseudocode here. A more complete version will be added in the final paper, and we commit to releasing the full codebase.
>
> ```
> # Input: current layer's hidden_states and QKV
> # Output: selected tokens for current layer
> # QKV shape: [B, N, D] — batch size, sequence length, embedding dim
> # If the architecture without CLS token, use a globally pooled token as a substitute
>
> cls_token = output.hidden_states[:,0] # [B,D]
> patch_token = output.hidden_states[:,1:] # [B,N-1,D]
>
> Q_cls = Q[:, 0]             # [B, D], first token is CLS token
> K_patch = K[:, 1:]          # [B, N-1, D], exclude CLS token
>
> # Compute CLS-to-token attention (O(N) complexity, compatible with FlashAttention)
> cls_attn = softmax(dot(Q_cls, K_patch.transpose(-1, -2)) / sqrt(D))  # [B, N-1]
>
> # Compute semantic similarity to evaluate token-level informativeness
> V_cls = V[:, 0]             # [B, D]
> V_patch = V[:, 1:]          # [B, N-1, D]
> semantic_attn = softmax(dot(V_cls, V_patch.transpose(-1, -2)) / sqrt(D))  # [B, N-1]
>
> # Compute value norm as a proxy for token information capacity
> v_score = l1_norm(V_patch)  # [B, N-1]
>
> # Fuse multiple importance criteria
> importance = (normalize(cls_attn) + normalize(semantic_attn)) * v_score  # [B, N-1]
>
> # Accumulate importance across layers (EMA-like smoothing)
> cumulative_score = 0.5 * cumulative_score + 0.5 * importance
>
> # Estimate pruning ratio based on CLS attention entropy
> entropy = compute_entropy(cls_attn)          # [B]
> entropy_ratio = entropy / log(N)
> prune_ratio = (N - target_num) / sqrt(remain_layer) * (1 - entropy_ratio ** 2)
>
> # Select top-K tokens based on cumulative importance scores
> keep_idx = topk(cumulative_score, K=round((1 - prune_ratio) * N))  # [B, K]
>
> # Retain cls token and selected tokens
> if not final_step: tokens = concat(cls_token, filter(patch_tokens, keep_idx))
> # Retain selected tokens
> elif final_step: tokens = filter(patch_tokens, keep_idx)
> ```
>
> ### 3. Retain less token (for weakness 3)
>
> Thank you for the valuable suggestion. To further validate our method under highly constrained token budgets, we conduct experiments with **only 32 visual tokens**. Our method outperforms existing methods by a notable margin under aggressive token reduction, validating its effectiveness in preserving informative content despite severe compression.
>
> |                            | Token |   GQA    | TextVQA  |  VQA-V2  |   POPE   |   SQA    |  VizWiz  |    Avg    |
> | :------------------------- | :---: | :------: | :------: | :------: | :------: | :------: | :------: | :-------: |
> | Llava-1.5 7B (upper limit) |  576  |   61.9   |   58.2   |   78.5   |   85.9   |   69.5   |   50.0   |   100%    |
> | FastV                      |  32   |   46.2   |   51.5   |   56.0   |   35.7   |   68.3   |   49.1   |   78.7%   |
> | VisionZip                  |  32   |   51.5   |   51.7   |   65.1   |   68.3   |   68.2   |   49.9   |   88.7%   |
> | Ours                       |  32   | **52.1** | **53.0** | **66.9** | **69.6** | **68.7** | **52.3** | **90.8%** |
>
> ### 4. Compare with CrossGET (for weakness 4)
>
> CrossGET injects learnable *cross tokens* (distinct from the CLS token) into each modality to capture cross-modal information, and evaluates token importance using cosine similarity with respect to these tokens, followed by merging less important tokens via complete-graph soft matching.
>
> Our method differs from CrossGET in three key aspects: **1) Theoretical Foundation:** CrossGET proposes a token reduction technique for inference acceleration. Beyond merely proposing a method, our work introduces a novel perspective (information flow) for understanding the emergence of redundant tokens. We find that the CLS token acts as an information relay and thus leverage it to simplify the modeling of token-to-token interaction, offering a comprehensive analysis of information flow to guide method design (as recognized by reviewer T8rC & NDVm). **2) Methodological Superiority:** CrossGET does not adopt the CLS token but injects learnable *cross tokens* to capture cross-modal information and utilizes *cross tokens* as a single criterion to assess token importance. In contrast, our method performs multi-layer and multi-criteria evaluation (CLS attention serves as one of the metrics), enabling more accurate identification of redundant tokens. We also introduce dynamic layer-wise pruning to ensure that pruning decisions align with the evolving information flow of tokens. **3) Training-Free Application:** CrossGET requires re-training due to its learnable *cross tokens*. In comparison, our method is entirely training-free while still achieving superior performance.
>
> As shown below, despite being training-free, our method reduces FLOPs to 50% and consistently outperforms CrossGET in accuracy and speed. Note that our method can achieve even better performance with training.
>
> |                              | FLOPs $\downarrow$ |   GQA    | TextVQA  |  VQA-V2  |   POPE   |   SQA    |  VizWiz  |
> | :--------------------------- | :----------------: | :------: | :------: | :------: | :------: | :------: | :------: |
> | Llava-1.5 7B (upper limit)   |        100%        |   61.9   |   58.2   |   78.5   |   85.9   |   69.5   |   50.0   |
> | CrossGET (requires training) |        66%         |   61.4   |   54.9   |   77.3   |   83.9   |   66.7   |   47.7   |
> | Ours (w/o train)             |      **50%**       | **61.5** | **57.8** | **77.6** | **86.1** | **68.7** | **52.1** |
> | Ours (w/ train)              |      **43%**       | **62.0** | **58.1** | **78.7** | **86.0** | **68.9** | **52.1** |

---

> > ### Comment · Reviewer_4FwU · 2025-08-04
> >
> > Thank the authors for providing a rebuttal. My concerns have been addressed, and I have raised my score. I'd suggest the authors carefully revise and include the new experiments and discussion to their final version.

---

> > > ### Author Response · Authors · 2025-08-04
> > >
> > > We sincerely thank you for acknowledging our response. We will carefully incorporate the new experiments and discussions into the final version

---

### Comment · Area_Chair_nRJB · 2025-08-04

Dear Reviewers,

If you have not done so already, please review the authors' rebuttal to your comments and the other reviews.  Please submit any further questions for the authors promptly to allow time for discussion.

Please also remember to update your ratings and final justification if your concerns have been addressed. If ratings are not updated, clearly explain any remaining concerns in your final justification.

As a reminder, the author-reviewer discussion period ends on August 6th, 11:59 PM AoE.

Best, Your AC

---

### Decision · Program_Chairs · 2025-09-17

**Decision:**

Accept (poster)

**Comment:**

The paper received three borderline accepts and one accept recommendations. The reviewers felt that the rebuttal addressed most of their initial concerns regarding some lack of experiments, background literature, and issues with clarity.  Based on the paper's good approach, final comprehensive experiments, and promising results, the ACs agree with the reviewers and recommend acceptance.  Please revise and include the new experiments and discussion in the final version.